# NVE-Adaptor: Novel View Editing Adaptor for Unseen View Consistent 3D Editing

## Abstract

3D editing aims to transform a given 3D structure according to the user's intent. Multi-view consistent 3D editing has been proposed to ensure consistent editing effects across different views of a 3D model, resulting in high-quality 3D structures. However, such consistency is only observed from viewpoints near the trained reference images, while renderings from other viewpoints (i.e., unseen view) often appear inconsistent and blurry. This is because current 3D editing systems are heavily optimized only for the given reference viewpoints. In real-world scenarios, it is also challenging for human to manually identify all regular viewpoints that ensure consistent 3D quality. To this end, we propose Novel View Editing Adapter (NVE-Adaptor), which enables 3D editing systems to maintain consistent quality even in unseen views. NVE-Adaptor supplements the limited reference views by sensibly exploring novel view in 3D space with rendered images from those views. These images are then refined using diffusion-based editing and used as additional supervision to improve view consistency. Our concept is simple, model-agnostic, and broadly applicable to multi-view 3D editing systems. We demonstrate its effectiveness on two 3D scene benchmarks (Mip-NeRF 360, Instruct-NeRF2NeRF) as well as on real-world data. The code will be made publicly available.

## 1 Introduction

Recent 3D reconstruction technologies Mildenhall et al. (2021); Kerbl et al. (2023) have significantly improved the quality of 3D outputs, and their integration Poole et al. (2022); Long et al. (2024) with diffusion models Dhariwal & Nichol (2021); Song et al. (2020b;a); Ho et al. (2020) is enabling the generation of more creative and diverse 3D content. These advancements are increasingly being applied in areas such as virtual reality, gaming, digital content creation, and even film production, where high-quality 3D assets are essential for immersive and interactive experiences. Building on recent advances in 3D generation, we focus on text-based 3D editing, where users can directly modify generated 3D content through text instructions to match their intended design.

Text-based 3D editing takes a 3D structure (*e.g.*, Neural Radiance Fields Mildenhall et al. (2021), 3D Gaussians Kerbl et al. (2023)) and a target text as input, and aims to produce a modified 3D structure aligned with the target text. Early approaches Lin et al. (2023); Haque et al. (2023); Sella et al. (2023); Zhuang et al. (2023) to 3D editing utilized score distillation sampling Poole et al. (2022) to update parameters of 3D reconstruction by computing gradients based on a score that measures the semantic alignment between the original 3D structure and the target text. Recent studies Chen et al. (2024); Wu et al. (2024) have introduced multi-view consistent 3D editing to improve the consistency of editing effects across different viewpoints of 3D structure. This multi-view 3D editing utilizes a set of reference images from multiple viewpoints that were used to construct the 3D structure, where text-based image editing Brooks et al. (2023) is applied to these reference images, and the 3D reconstruction is then fine-tuned based on the edited images, leading to the generation of the modified 3D structure. These edited reference images enable 3D structures to maintain consistent editing effects across different viewpoints, and several editing consistency modules Geyer et al. (2023); Liu et al. (2024) have been applied to achieve this. Although multi-view 3D editing systems maintain high consistency of editing effect and quality near the reference image views, they still suffer from inconsistency in 3D editing due to quality degradation in areas from unseen viewpoints that were not trained in the fine-tuning process.

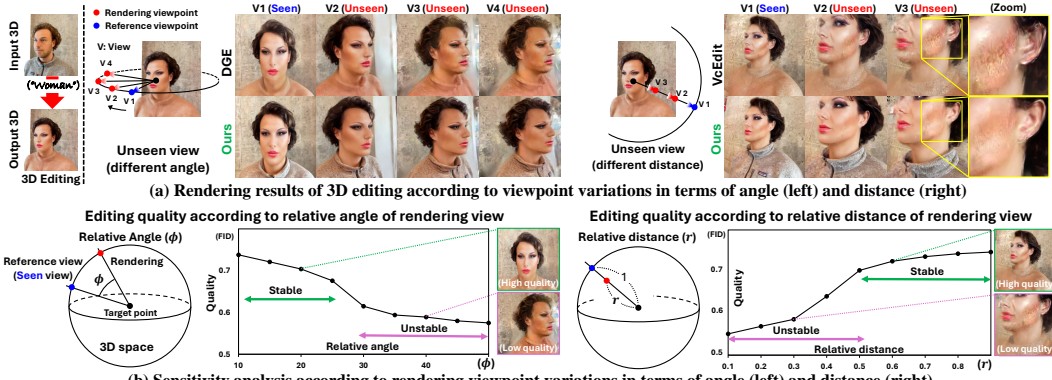

(a) Rendering results of 3D editing according to viewpoint variations in terms of angle (left) and distance (right)

(b) Sensitivity analysis according to rendering viewpoint variations in terms of angle (left) and distance (right)

Figure 1: Illustration of vulnerability of unseen view rendering results of current 3D editing systems Chen et al. (2024), Wang et al. (2024). (a) shows the rendering results of 3D editing from various unseen views, exhibiting that quality degrades as the viewpoint moves farther from the trained reference view. (b) presents a sensitivity analysis of 3D editing quality according to different viewpoints, showing that the degradation emerges when the rendering viewpoint (red) deviates beyond a certain angle (*e.g.*, $\phi > 25°$) or distance (*e.g.*, $r < 0.5$) from the reference view (blue).

Figure 1 (a) shows current 3D editing results rendered from unseen viewpoints. On the far left, the edited 3D output is depicted, where to explore unseen viewpoints in the output, we investigate the rendering results of the 3D structure by gradually diverging from the trained viewpoint (reference view) for 3D editing. First, we examined a spherical region centered around a target point (black as the mean point of the 3D structure), maintaining the same distance to the reference viewpoint (blue). It was observed that the quality of the 3D editing deteriorates as the viewpoint (red) moves farther away from the reference viewpoint. Additionally, we evaluated the quality from other unseen viewpoints located at internal division points along the line connecting the reference viewpoint and the target point, observing the target from those positions. As the viewpoint for rendering moves closer to the target, the results become increasingly blurry. Motivated by these observations, we conducted a sensitivity analysis with respect to changes in the location of unseen viewpoints. Specifically, the left of Figure 1 (b) shows our study of a spherical region that maintains the same distance from the target to the reference point. To analyze this region, we define a relative angle $\phi$ among unseen rendering viewpoint, target point, and reference point. By varying $\phi$, we evaluated how the quality of rendered images changes across different viewpoints within this spherical region. We observed that when the unseen viewpoint deviates more than approximately 25 degrees from the reference, the quality of the 3D structure becomes unstable[1]. The right side of Figure 1 (b) shows the exploration of unseen views that gradually approach the target point from the reference viewpoint. Assuming the distance from the target to the reference is normalized to 1, and defining the position of the unseen viewpoint by the ratio $r$, we observed that unstable quality began to appear when $r$ dropped below approximately 0.5. This indicates that, in order to achieve consistent quality in 3D editing (or generation) across different viewpoints, there must always be a reference view available in the vicinity[2] of any given viewpoint. However, in real environments, when humans manually capture 3D images to provide rendering supervision, the image composition tends to be highly non-uniform. Conversely, capturing video to scan the entire spatial region provides excessively redundant reference views, which dramatically increases training time and resources, resulting in inefficient training of 3D editing.

To address these issues, we propose a novel view editing adaptor (NVE-Adaptor), a simple and intuitive method to reduce view inconsistency. As shown in Figure 2 (a), conventional multi-view consistent 3D editing begins by editing the reference image using a diffusion model to generate an edited reference. The 3D structure is then fine-tuned based on this edited reference to perform the 3D editing. However, reference images are limited and often fail to cover enough viewpoints. NVE-Adaptor supplements the limited reference views by exploring novel viewpoints in 3D space with

---

[1]To avoid the situation where the viewpoint becomes closer to another reference viewpoint due to excessive angular deviation, the analysis was limited to angles below approximately 50 degrees.

[2]Although "vicinity" is somewhat ambiguous, we found that 3D editing quality is preserved when a viewpoint lies within $\phi < 25$ or $0.5 < r < 1$ of the nearest reference. This is being conducted for motivation, and of course, our proposed method does not rely on these empirical observations.

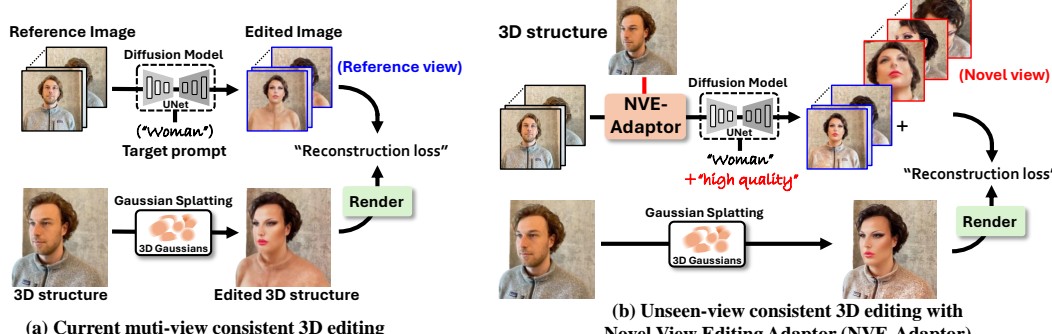

**(a) Current muti-view consistent 3D editing**

**(b) Unseen-view consistent 3D editing with Novel View Editing Adaptor (NVE-Adaptor)**

Figure 2: An overview of (a) the conventional multi-view consistent 3D editing framework and (b) our proposed Novel View Editing Adaptor (NVE-Adaptor) for unseen-view consistent 3D editing. Existing methods edit the reference view (used to build the initial 3D structure) via a diffusion model, and use the result as supervision for 3D editing. NVE-Adaptor targets quality degradation in unseen views by exploring novel viewpoints from 3D structure, editing their renderings, and provides edited novel reference as additional supervision for multi-view 3D editing systems.

rendered images from those views. Since, this approach depends on a proper quality of renderings at novel viewpoints, we employ a diffusion model to enhance the quality of novel view renderings by editing them[3]. The edited novel views serve as additional supervision to improve overall editing consistency. To further enhance this process, we input novel view renderings along with actual captured reference images into diffusion editing to share enhanced quality of reference images via the diffusion consistency module (*e.g.*, attention, propagation), while including explicit instructions such as "high quality" in the prompt to actively boost rendering fidelity. The NVE-Adaptor can be seamlessly integrated into any multi-view 3D editing system in a model-agnostic manner, improving unseen view quality. Extensive experiments on public 3D scene datasets (Mip-NeRF 360 Barron et al. (2021) and Instruct-NeRF2NeRF Haque et al. (2023)) show consistent improvements on both seen and unseen views across various 3D editing baselines. Furthermore, we validate the effectiveness on an additional real-world test set, demonstrating robustness even on challenging unseen views.

## 2 RELATED WORK

### 2.1 TEXT-BASED 3D EDITING

Text-based 3D editing has gained attention for enabling 3D object and scene generation or modification through text prompts. Advances in 3D reconstruction methods like Neural Radiance Fields (NeRF) Mildenhall et al. (2021) and Gaussian Splatting Kerbl et al. (2023) have improved fidelity and speed, forming a strong basis for 3D generation. Building on the success of diffusion models such as Stable Diffusion Rombach et al. (2022), these methods have enabled text-driven generation of 3D shapes, expanding the flexibility of content creation. For instance, DreamFusion Poole et al. (2022) and Magic3D Lin et al. (2023) introduced score distillation sampling to optimize NeRF representations from textual descriptions, yielding high-fidelity 3D outputs with minimal manual effort. More recent work, such as Instruct-NeRF2NeRF Haque et al. (2023), allows direct editing of existing 3D models via natural language. Across these endeavors, researchers have tackled key challenges of effective editing using attention control Luo et al. (2024), dividing sub-tasks Chen & Wang (2024), and leveraging latent without unintended changes Khalid et al. (2024). Recently, many works are focusing on multi-view consistency Dong & Wang (2023); Chen et al. (2024); Wu et al. (2024); Wang et al. (2024). Despite notable progress, handling unseen views remains a critical challenge in text-based 3D editing, motivating us to focus on robust solutions for unseen view consistency while maintaining high-quality results and user-driven modifications.

---

[3]Depending on the application, our concept can improve unseen-view quality in both 3D editing and 3D reconstruction. For example, in reconstruction, low-quality rendering images of unseen views are enhanced by a diffusion model and then use its outputs as pseudo ground-truth for reconstruction. Empirically, however, it is more effective for 3D editing, since latent diffusion models are trained to generate semantic content rather than to enhance fidelity without semantic change. We therefore highlight text-based 3D editing as the primary focus of our method, while additional results on reconstruction are also provided in Figure 8.

## 2.2 Novel View Synthesis

Novel view synthesis has advanced through improved scene representations and rendering techniques, from traditional geometry-based methods Seitz & Dyer (1996); Shade et al. (1998) and light fields Levoy & Hanrahan (2023) to learning-based approaches of multi-plane images Zhou et al. (2018); Penner & Zhang (2017) and volumetric neural representations, notably Neural Radiance Fields (NeRF) Mildenhall et al. (2021) and its variants Barron et al. (2021); Reiser et al. (2021). Recent innovations such as Gaussian Splatting Kerbl et al. (2023); Yang et al. (2024); Lu et al. (2024) replace neural volumetric fields with a set of 3D Gaussians, enabling real-time novel view synthesis through direct rasterization and alpha blending. Despite these extensive efforts to boost overall quality, a persistent gap remains between the regions well-sampled during training and genuinely unseen viewpoints. In this work, we propose the first approach to tackle this limitation in reverse: leveraging generative editing capabilities (via diffusion models) to enhance the rendered images from novel views and then reusing these improved images as supervision to refine the underlying 3D structure. By doing so, we aim to mitigate the quality deterioration often observed in novel views.

## 3 Preliminary

### 3.1 Radiance Fields for View Rendering

A *radiance field* is defined as a pair of mapping functions $\sigma : \mathbb{R}^3 \to \mathbb{R}^+$ and $c : \mathbb{R}^3 \times \mathbb{S}^2 \to \mathbb{R}^3$, where $\sigma(\mathbf{x})$ represents the volumetric density (or opacity) at a 3D point $\mathbf{x} \in \mathbb{R}^3$, and $c(\mathbf{x}, \boldsymbol{\nu})$ denotes the color radiance emitted from $\mathbf{x}$ in the viewing direction $\boldsymbol{\nu}(\theta, \phi) \in \mathbb{S}^2$ of 3D unit sphere ($\theta, \phi$ are polar and azimuthal angle). An image $I$ observed from this radiance field is computed using the emission-absorption equation Levoy (2002), which models light accumulation along a camera ray as:

$$I(\mathbf{u}) = \int_0^\infty c(\mathbf{x}_t, \boldsymbol{\nu}) \, \sigma(\mathbf{x}_t) \exp\left( - \int_0^t \sigma(\mathbf{x}_\tau) \, d\tau \right) dt, \tag{1}$$

where the 3D point $\mathbf{x}_t = \mathbf{x}_0 - t\boldsymbol{\nu}$ parameterizes the ray with $t$, originating from the camera center $\mathbf{x}_0$ (in world coordinate) and passing through the pixel $\mathbf{u}$ in the direction $-\boldsymbol{\nu}$. Previous works have explored various representations for these functions, including voxel grids Sun et al. (2022); Chan et al. (2022) and multilayer perceptrons (*i.e.*, NeRF Mildenhall et al. (2021)). A recent approach, Gaussian Splatting Kerbl et al. (2023), represents the scene as a mixture of 3D Gaussians defined as $\mathcal{G} = \{(\sigma_i, \mu_i, \Sigma_i, c_i)\}_{i=1}^G$. Here, for $i$-th Gaussian, $\sigma_i \geq 0$ denotes the opacity, $\mu_i \in \mathbb{R}^3$ is the mean position, $\Sigma_i \in \mathbb{R}^{3 \times 3}$ is the covariance matrix defining the spatial extent and orientation, and $c_i : \mathbb{S}^2 \to \mathbb{R}^3$ specifies the directional color emitted in direction $\boldsymbol{\nu} \in \mathbb{S}^2$. Specifically, for representing multiple colors according to the view direction, the color is designed by combination of spherical harmonics Fridovich-Keil et al. (2022) as $c_i(\nu) = \sum_{l=0}^L \sum_{m=-l}^l c_{ilm} Y_{lm}(\nu)$, where $Y_{lm}$ are spherical harmonic basis and $c_{ilm} \in \mathbb{R}$ are corresponding coefficients with max degree $L$ affecting the number of harmonics. The Gaussian function is formulated as $g_i(x) = \exp\left( -\frac{1}{2} (x - \mu_i)^\top \Sigma_i^{-1} (x - \mu_i) \right)$. Thus, the mixture of Gaussians is constructed by the opacity and color functions as given below:

$$\sigma(x) = \sum_{i=1}^G \sigma_i g_i(x), \quad c(x, \nu) = \frac{\sum_{i=1}^G c_i(\nu) \sigma_i g_i(x)}{\sum_{i=1}^G \sigma_i g_i(x)}. \tag{2}$$

Notably, this compact and continuous nature of the Gaussian representation allows for high-fidelity rendering while maintaining a relatively low memory and high speed. Above all, our method can be applied to 3D editing systems that use a variety of radiance field representations.

## 4 Method

In this section, we first provide a conceptual explanation of how multi-view 3D editing systems work. We then propose Novel View Editing Adaptor (NVE-Adaptor) as a way to improve the robustness of 3D editing quality on unseen views. To make the distinction clear between 'novel' and 'unseen', our method explores unseen views and refers to the selected views among them as novel views.

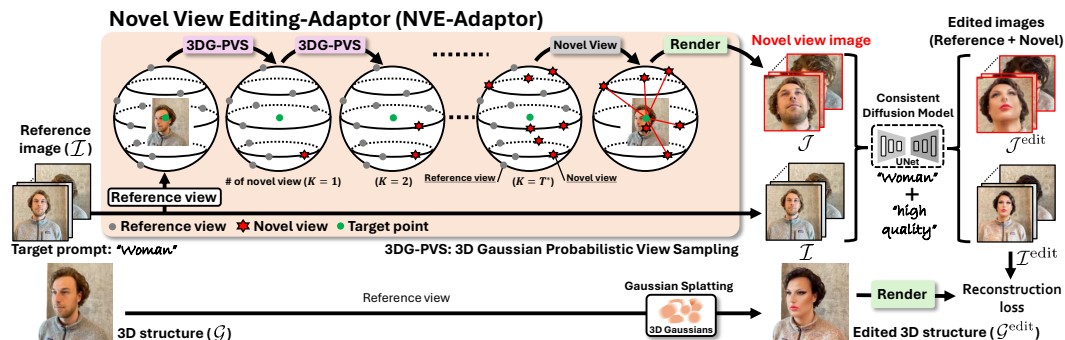

Figure 3: Novel View Editing Adaptor (NVE-Adaptor) takes camera viewpoints (reference viewpoints) for capturing reference images and explores novel viewpoints in regions spatially distant from them. To enable effective exploration, we introduce 3D Gaussian Probabilistic View Sampling (3DG-PVS), which iteratively samples novel views based on the reference and previously chosen viewpoints. Sampled novel view images and reference images are jointly processed by diffusion model with consistency modules (*e.g.*, attention, propagation) for multiple images and further guided by an additional instruction (*e.g.*, "high quality") to enhance the rendering quality of the novel views. The edited novel-view images then serve as additional supervision for the 3D editing.

## 4.1 Novel View Editing Adaptor

Given a mixture of 3D Gaussians[4] $\mathcal{G}$ and target textual prompt $\mathcal{T}$, 3D editing aims to provide 3D structure $\mathcal{G}^{\text{edit}}$ that conforms to the target prompt. For this 3D editing, multi-view 3D editing systems have been an intuitive approach to improve view consistency during the editing optimizations of 3D generation models (*e.g.*, Gaussian Splatting, NeRF) toward a desired target. They use reference multi-view images $\mathcal{I} = (I_t)_{t=1}^{T}$ captured from $T$ camera viewpoints $(\pi_t)_{t=1}^{T}$, which were originally used to train for reconstructing the input 3D structures, and apply text-based diffusion editing (*e.g.*, InstructPix2Pix Brooks et al. (2023)) to them to produce edited images $\mathcal{I}^{\text{edit}}$ that conform to the target text. These systems then optimize [5] the Gaussian $\mathcal{G}$ so that its renderings align with edited images as:

$$\mathcal{G}^{\text{edit}} = \arg\min_{\mathcal{G}} \sum_{t=1}^{T} \left\| I_t^{\text{edit}} - \text{Render}(\mathcal{G}, \pi_t) \right\|, \tag{3}$$

where the $I_t^{\text{edit}}$ is edited image and $\text{Render}(\mathcal{G}, \pi_t)$ is 2D rendering of $\mathcal{G}$ from a $t$-th viewpoint $\pi_t$. This multi-view 3D editing's training objective ensures high-quality 3D structures around the reference viewpoints and their nearby views. However, in real-world scenarios, the actual viewpoints provided by users are often unevenly distributed, making them sparser and vulnerable to many unseen views. Therefore, the proposed NVE-Adaptor is designed to address the vulnerability of quality degradation across unseen viewpoints, and its integration is performed with a simple modification of the training objective of multi-view 3D editing. Figure 3 presents an overview of NVE-Adaptor. It samples novel viewpoints $\pi^*$ away from reference view points $\pi$, where we introduces another novel view guidance term in the training objective with approximated edited reference image $\mathcal{J}^{\text{edit}} = (J_t^{\text{edit}})_{t=1}^{T^*}$ as below:

$$\mathcal{G}^{\text{edit}} = \arg\min_{\mathcal{G}} \left( \sum_{t=1}^{T} \left\| I_t^{\text{edit}} - \text{Render}(\mathcal{G}, \pi_t) \right\| + \alpha \sum_{t=1}^{T^*} \left\| J_t^{\text{edit}} - \text{Render}(\mathcal{G}, \pi_t^*) \right\| \right), \tag{4}$$

where the $T^*$ is the number of sampled novel viewpoints and $\alpha$ is scaler $(T/T^*)$ to balance the impacts of reference and novel views. To ensure the effectiveness of the novel view guidance term, it is crucial to supply high-quality $J_t^{\text{edit}}$ as supervision. To this end, we first render low-quality images $\mathcal{J}$ from each novel viewpoint $\pi_t^*$ and subsequently enhance their quality through text-based diffusion editing. To be specific, we perform frame consistent editing by inputting the novel view images and the reference images together as $[\mathcal{I}^{\text{edit}}, \mathcal{J}^{\text{edit}}] = \text{Editor}([\mathcal{I}, \mathcal{J}], \mathcal{T})$. This encourages the high quality of the reference image $\mathcal{I}$ to be shared to $\mathcal{J}$ through the consistency module[6] (*e.g.*, attention, propagation). In addition, to maximize the text-based editing capability, we add target prompt with explicit instruction (*e.g.*, 'high quality')[7] that requires quality improvement. While ensuring high quality in $\mathcal{J}^{\text{edit}}$ is important, it is also crucial to sufficiently and uniformly explore space of potential effective novel viewpoints $\pi^*$. Therefore, we propose a novel view sampling strategy in the following.

---

[4]We represent 3D structure with 3D Gaussians, but our method is also available of other forms (*e.g.*, NeRF).

[5]Each baseline may have a slightly different training objective, where we adaptively follow it accordingly.

[6]To confirm the effectiveness of our method, we follow the same consistency module for each editing system.

[7]Various prompts for quality enhancement are also available, with analyses provided in Appendix E.

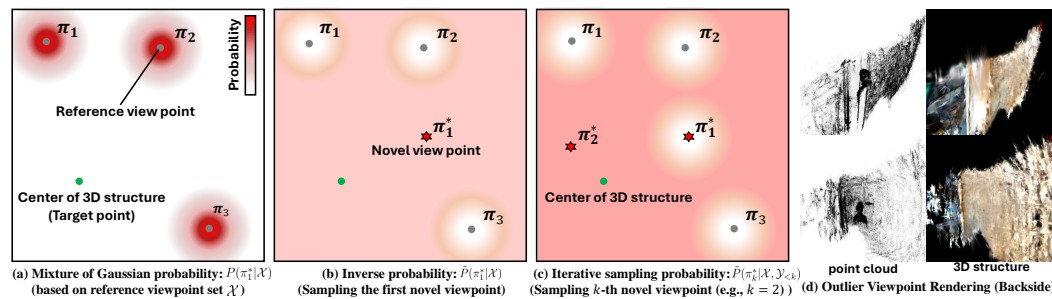

Figure 4: Visualization of probability distribution for sampling novel viewpoint: (a) probability of mixture 3D Gaussians, (b) its inverse probability for sampling the first novel viewpoint, (c) iterative probability for the $k$-th sampling novel viewpoint, (d) examples of outlier viewpoint to be removed.

## 4.2 3D GAUSSIAN PROBABILISTIC VIEW SAMPLING

Our intended novel viewpoints should be sampled with diversity to traverse many unseen views and be effective to be fully beyond the original reference viewpoints set $\{\pi_t\}_{t=1}^T \in \mathcal{X}$. To this end, we design 3D Gaussian Probabilistic View Sampling (3DG-PVS), which enables the iterative acquisition of diverse novel viewpoint set $\{\pi_t^*\}_{t=1}^{T^*} \in \mathcal{Y}$ in 3D space. Since the sampling process is iterative, we describe it using a recursive formulation. We define the sampling probability of the first novel viewpoint sampling as $P(\pi_1^*|\mathcal{X})$ and then generalized to the $k$-th sampling step as $P(\pi_k^*|\mathcal{X}, \mathcal{Y}_{<k})$ conditioned on the reference set $\mathcal{X}$ and all previously selected novel viewpoints as $\{\pi_t^*\}_{t=1}^{k-1} \in \mathcal{Y}_{<k}$. Specifically, for the $P(\pi_1^*|\mathcal{X})$, we intend to sample viewpoints from regions distant from the viewpoints set $\mathcal{X}$ by employing a probability distribution from a mixture of isotropic 3D Gaussians centered at the reference viewpoints, as illustrated in Figure 4 (a). Sampling is then performed based on the inverse probability $\tilde{P}$ of Figure 4 (b), encouraging selection of viewpoints away from the reference set as defined below:

$$P(\pi_1^*|\mathcal{X}) = \frac{1}{T}\sum_{t=1}^{T}\mathcal{N}(\pi_1^*|\pi_t, \Sigma_t), \quad \tilde{P}(\pi_1^*|\mathcal{X}) = \frac{1 - P(\pi_1^*|\mathcal{X})}{Z}, \quad Z = \int (1 - P(\pi_1^*|\mathcal{X}))d\pi_1^*, \quad (5)$$

where $\Sigma_t = \sigma^2 I \in \mathbb{R}^{3\times 3}$ is an Isotropic 3D covariance with $\sigma = 0.4$ and $Z$ is normalization scaler. For the $k$-th sampling $\{\pi_t^*\}_{t=1}^{k-1} \in \mathcal{Y}_{<k}$, we simply update[8] the mixture of 3D Gaussians together with sampled novel viewpoint set $\mathcal{Y}_{<k}$. Thus, this iterative formulation prevents the sampled novel viewpoints from clustering. However, as shown in Figure 4 (d), there are regions (*e.g.*, back side of the scene) that do not require to sample. To handle this, we define outlier removal as exclusion space using a plane that marks space to avoid during sampling (also available with algorithm in Appendix C). If a sampled viewpoint falls within this excluded space, it is discarded and resampled. Once $T^*$ novel view samples are obtained, they are applied into Equation (4). At the end of each training epoch, a new set of novel views is sampled again to encourage diverse and robust supervision.

## 5 EXPERIMENT

### 5.1 EXPERIMENTAL SETTINGS

**Implementation Details.** For 3D generation, we use Gaussian Splatting Kerbl et al. (2023) and NeRF Mildenhall et al. (2021), with image editing using Stable Diffusion 1.5 Rombach et al. (2022). The number $T^*$ of novel views is set to 30 from Table 2 and the number of reference view is given by $60 < T < 180$. We follow baselines' image and text encoders on two NVIDIA A100 GPUs.

**Data and Baselines.** We use multiple scenes from Instruct-NeRF2NeRF (IN2N) Haque et al. (2023) and Mip-NeRF 360 Barron et al. (2022) datasets for evaluation and created a separate validation set for ablation study. To test unseen view quality, we constructed ground-truth renderings using real photographs captured from 14 real-world scenes as 'unseen view set', and assessed the quality of

---

[8]Appendix C provides precise definition and detailed iterative sampling algorithm.

Table 1: Quantitative evaluations of NVE-Adaptor with 3D editing systems on IN2N and Mip-NeRF 360 datasets (Evaluations on 95% confidence interval, dataset-specific analysis and scene-wise analysis are given in Appendix E). $CLIP_T$ is text-to-image clip for textual alignment, $CLIP_D$ is directional clip score, and $CLIP_I$ is image-to-image clip score for consistency. It is reported in a format of (original test set / unseen view set).

| Method | Consistency | | Fidelity | | Textual Alignment | | Human |
|---|---|---|---|---|---|---|---|
| | FID ↓ | $CLIP_I$ ↑ | LPIPS ↓ | SSIM ↑ | $CLIP_T$ ↑ | $CLIP_D$ ↑ | |
| IN2N Haque et al. (2023) | 256/307 | 0.904/0.870 | 0.377/0.442 | 0.623/0.576 | 0.198/0.183 | 0.048/0.035 | 0.22 |
| IN2N + NVE-Adaptor | 230/261 | 0.913/0.901 | 0.330/0.387 | 0.666/0.617 | 0.208/0.193 | 0.059/0.047 | 0.78 |
| GCtrl Wu et al. (2024) | 245/294 | 0.906/0.872 | 0.356/0.423 | 0.637/0.583 | 0.201/0.186 | 0.056/0.039 | 0.25 |
| GCtrl + NVE-Adaptor | 217/248 | 0.915/0.902 | 0.311/0.362 | 0.672/0.621 | 0.210/0.198 | 0.064/0.055 | 0.75 |
| VcEdit Wang et al. (2024) | 232/284 | 0.913/0.878 | 0.338/0.417 | 0.654/0.606 | 0.209/0.185 | 0.062/0.047 | 0.34 |
| VcEdit + NVE-Adaptor | 209/240 | 0.923/0.909 | 0.300/0.345 | 0.694/0.647 | 0.218/0.206 | 0.068/0.060 | 0.66 |
| DGE Chen et al. (2024) | 223/271 | 0.921/0.886 | 0.329/0.399 | 0.670/0.614 | 0.208/0.190 | 0.062/0.047 | 0.33 |
| DGE + NVE-Adaptor | 201/233 | 0.929/0.917 | 0.294/0.336 | 0.704/0.656 | 0.215/0.206 | 0.069/0.061 | 0.67 |

the 3D renderings against them, where extrinsic parameters (camera rotation and translation) for rendering can be extracted by COLMAP Schonberger & Frahm (2016). For the unseen view set, shown in Figure 1, we select 20 test viewpoints from unstable region about relative angle ($\phi > 25°$) and distance ($r < 0.5$) based on pre-defined 80 reference viewpoints. NVE-Adaptor is validated on recent editing systems with different approaches (Gaussian Splatting, NeRF) including IN2N, GaussCtrl (GCtrl) Wu et al. (2024), DGE Chen et al. (2024), VcEdit Wang et al. (2024).

## 5.2 EVALUATION METRICS

To evaluate the edited 3D structures, we leverage various metrics used for image and video editings. To assess how well the edited results align with the target text, we evaluate text-to-image CLIP Radford et al. (2021) score. Additionally, to measure editing responsiveness, we measure directional CLIP score, which quantifies whether the change from the original 3D's rendering to the edited 3D's rendering correctly reflects the transformation required by the text (*i.e.*, from source to target text) in terms of cosine similarity of the latent embedding differences. For the evaluation of fidelity to the input 3D structures, we measure Structural Similarity Index Measure (SSIM) Wang et al. (2004) and Learned Perceptual Image Patch Similarity (LPIPS) Zhang et al. (2018). To evaluate the consistency of 3D structure, we measure image-to-image CLIP score between consecutively rendered images along a camera path and measure Fréchet Inception Distance (FID) Heusel et al. (2017) between the rendered images and the actual captured ones. We also investigate human preferences on results from the baselines with and without NVE-Adaptor. All metrics are average scores over 10 random seeds.

## 5.3 EXPERIMENTAL RESULTS

**Quantitative Comparisons.** Table 1 summarizes the results of existing baselines on the original test set and novel view test set with and without applying the NVE-Adaptor to them. The novel view set evaluates the 3D editing results from unseen viewpoints, which were not included during the 3D editing training process. All models show lower performance on the unseen view set compared to the original test set, highlighting the challenges in maintaining quality for unseen views. For this unseen view set, when the NVE-Adaptor is applied to the baselines, it improves quality across all aspects, with particularly significant gains in consistency and fidelity. Furthermore, while we expected the NVE-Adaptor to only mitigate vulnerabilities on the unseen view set, it also showed performance improvements on the original test set composed of seen views. We consider this is because the additional rendered images from novel viewpoints $\pi^*$ serve as a bridge between seen viewpoints, improving visual correspondence and leading to more consistent quality in the generated outputs.

**Qualitative Comparisons.** Figure 5 qualitatively demonstrates the impact of the NVE-Adaptor across models. We sampled viewpoints near seen views used during training and additional challenging viewpoints by adjusting the relative angle ($\phi > 25°$) and distance ($r < 0.5$) to explore unseen regions. From the seen views, both the baseline and NVE-adapted one produce comparable results, but the latter shows noticeable improvement in background fidelity (yellow box), due to increased multi-view consistency provided by the adaptor's diverse views. In unseen views, the adaptor's effect becomes more prominent. For example, when changing the viewing angle to look upward, the baseline model exhibits needle-like noise and shape collapse, whereas the NVE-enhanced version

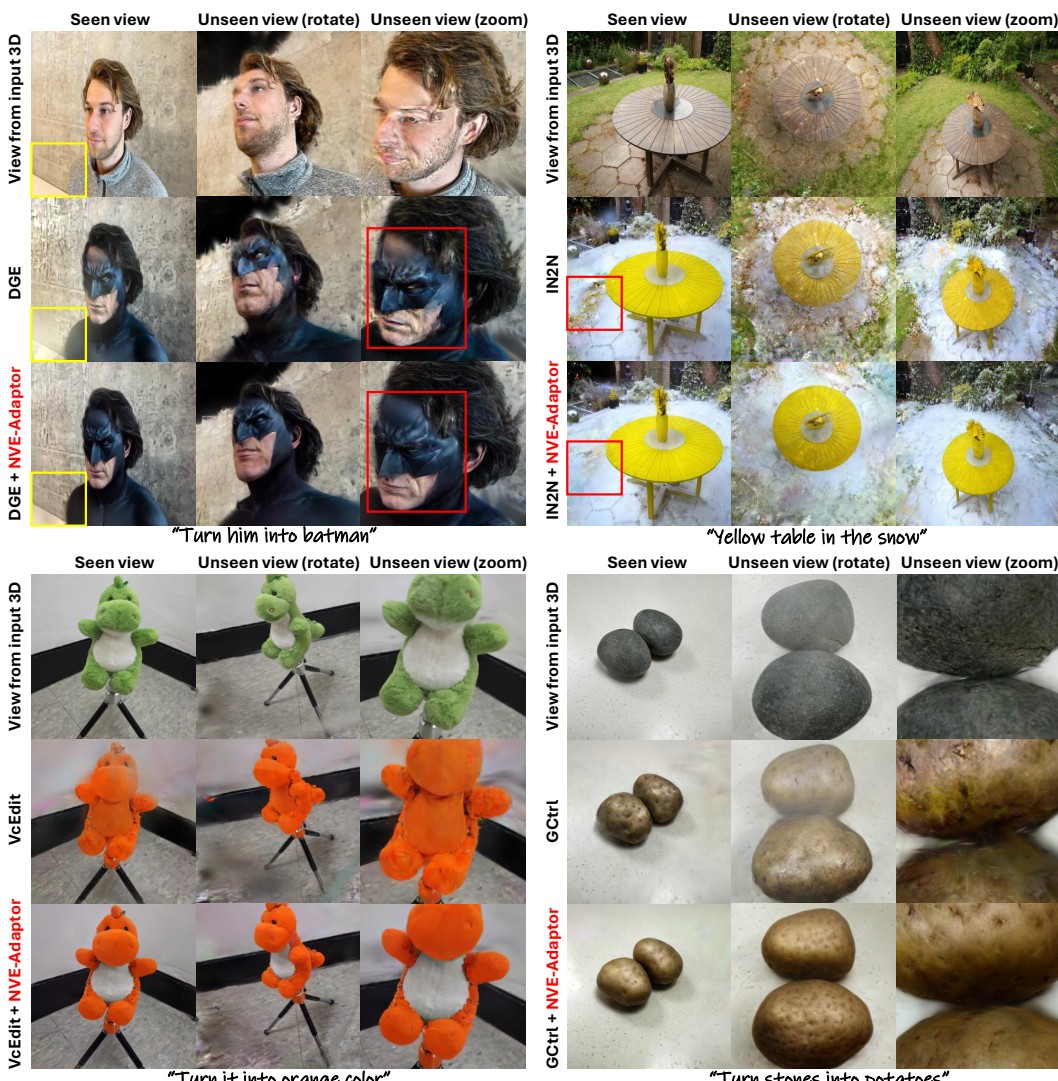

Figure 5: Qualitative results on 3D editing baselines with and without NVE-Adaptor. The first column of each sample shows the results from seen viewpoints and the second and the third row show rendering results from unseen view using rotation and zoom. The yellow box shows fidelity and the red shows editing effects. Our novel view images from NVE-Adaptor are also given in Appendix F.

maintains clarity (red box). Similarly, when zooming in, baseline models often fail to preserve the editing effect due to limited supervision in those regions, leading to uneven or absent edits. This pattern recurs in the second sample (right side of Figure 5), where edited region into a white snow results in inconsistent geometry and texture when viewed top-down or close-up. Applying our adaptor resolves these issues, yielding stable and consistent edits. Finally, the two examples at the bottom of Figure 5 are drawn from the novel view set used for quantitative evaluation (Table 1), where we confirm the NVE-Adaptor's consistent effectiveness across diverse novel view conditions.

**Robustness Analysis.** Figure 6 shows the relative angles and distances between 500 randomly (in pre-defined cube space, not to be far from the 3D structure) distributed viewpoints in 3D space and their nearest reference viewpoints (including novel viewpoints when integrated NVE-Adaptor). We observed that the higher number of viewpoints (in ratio) get to be positioned within stable ranges of both angle and distance after applying our NVE-Adaptor.

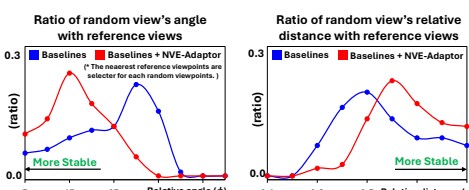

Figure 6: The relative angles and distances between 500 randomly scattered points in 3D space and their nearest reference viewpoints.

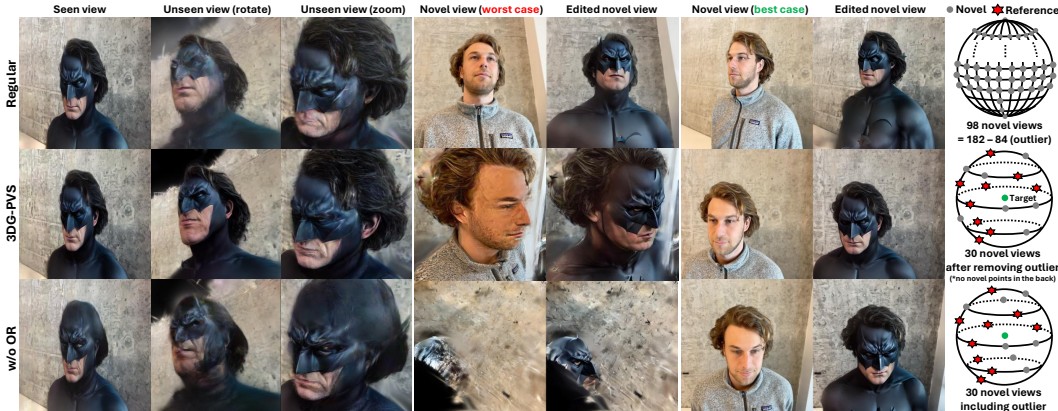

Figure 7: Qualitative results about ablation of novel view sampling methods. 'Regular' denotes the sampling of fixed points in 3D space and 'OR' denotes outlier removal. DGE is used for baseline.

Table 2: Ablation study of NVE-Adaptor (baselines' average score, Figure 7 explains 'Regular' and 'OR'. The 'iterative' means sampling. $T^*$: number of novel viewpoints).

| Method | Reference view | | Novel view | |
|---|---|---|---|---|
| | FID↓ | SSIM↑ | FID↓ | SSIM↑ |
| Regular | 217 | 0.688 | 264 | 0.610 |
| 3DG-PVS | **212** | **0.693** | **230** | 0.651 |
| w/o iterative | 225 | 0.671 | 279 | 0.604 |
| w/o OR | 265 | 0.622 | 304 | 0.584 |
| $T^* = 20$ | 229 | 0.683 | 246 | 0.626 |
| $T^* = 30$ | **212** | 0.693 | **230** | 0.651 |
| $T^* = 40$ | 214 | **0.695** | 235 | 0.648 |
| $T^* = 50$ | 213 | 0.694 | 232 | 0.644 |

Table 3: Training time of 100 frames with NVE-Adaptor (NVE).

| Method | Time |
|---|---|
| IN2N | 56 min |
| IN2N + NVE | 68 min |
| GCtrl | 23 min |
| GCtrl + NVE | 28 min |
| VcEdit | 21 min |
| VcEdit + NVE | 26 min |
| DGE | 15 min |
| DGE + NVE | 18 min |

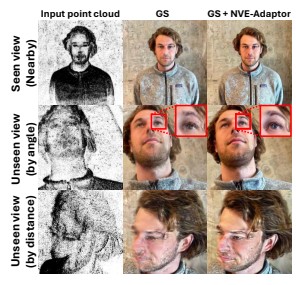

Figure 8: 3D generation with NVE-Adaptor (GS: Gaussian Splatting).

**Ablation Study and Computational Complexity Analysis.** Figure 7 and Table 2 present the ablation study of the NVE-Adaptor. We first investigate different strategies for novel view exploration. We compared our method with the most intuitive approach of uniformly distributing viewpoints in space regardless of the reference view locations as shown in the top right of Figure 7. While this method also showed some effectiveness, it was not as strong as our 3DG-PVS. Even if it requires a large number of novel viewpoints to achieve sufficient coverage, and the optimization is limited to a fixed set of predefined views throughout training. This restricts the model's ability to explore a broader range of novel viewpoints dynamically. When iterative sampling was removed, spatial redundancy among novel viewpoints increased, reducing overall effectiveness. Additionally, omitting outlier removal led to the inclusion of undesirable novel viewpoints (shown in Figure 7), which significantly degraded performance. Table 3 shows the time required to perform 3D editing on a single scene. Since our method introduces additional novel views, it incurs more computational time. While this overhead is significant in NeRF-based method, relatively low in Gaussian Splatting.

**Plug-and-Play into 3D reconstruction.** Since our adaptor can also be applied to 3D reconstruction, Figure 8 presents results from integrating it into a 3D generation. It shows improved rendering quality at novel viewpoints compared to the baseline. This is available because the diffusion model allows for edits that enhance image quality, but it has the drawback that the reconstruction occurs in two stages.

## 6 CONCLUSION

This paper introduces a Novel View Editing Adaptor (NVE-Adaptor) to enhance consistency in unseen views within 3D editing systems. Current methods focus on reference viewpoints used for 3D synthesis, which often causes quality degradation when rendering unseen perspectives. NVE-Adaptor addresses this by sampling and editing novel viewpoints away from the references, providing these edited renderings as additional supervision to improve overall view consistency. NVE-Adaptor enables simple integration with diverse 3D editing systems, effectively enhancing output quality.

## ETHICS STATEMENT

Our work explores 3D generative editing, which enables semantic manipulation of 3D content. While beneficial for applications like AR/VR and design, it also raises ethical concerns such as potential misuse for deceptive content, privacy risks, and bias from training data. As our method builds on generative models, it inherits these risks. To mitigate them, we will release our code and data details under a responsible-use license and are exploring techniques like 3D watermarking and digital forensics. We are committed to transparency and ethical deployment of 3D editing technologies.

## REPRODUCIBILITY STATEMENT

We provide the implementation details of our proposed NVE-Adaptor, including the algorithm for 3D Gaussian Probabilistic View Sampling, in Appendix C, where we also describe Outlier Removal and additional technical details. Moreover, our supplementary material contains the complete source code used in our experiments. The code is being prepared for public release to facilitate reproducibility and future research.

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

## A  THE USE OF LARGE LANGUAGE MODELS (LLMS)

In this paper, LLMs were used exclusively for polishing the writing and improving readability. They played no role in the conception of ideas or the development of the proposed model.

## B  LIMITATIONS

Text-based 3D editing inherently inherits the general limitations of text-driven editing. First, ambiguity in natural language can cause the intended edit to be misinterpreted or only partially reflected. Second, pre-trained generative models such as diffusion models often have limited understanding of text semantics, which makes it difficult for them to accurately localize the intended editing regions. As a result, unintended areas may be modified, or desired regions may not be edited correctly. We also want to provide our method's limitations in two perspective. First, it introduces additional computational overhead. To provide editing supervision from novel views, the system must render and edit additional viewpoints, which then require the 3D model to undergo further optimization. Second, when applied to 3D reconstruction tasks, NVE-Adaptor follows a two-stage process. Since edited novel views depend on an existing 3D model, methods such as Gaussian Splatting or NeRF must first complete a full reconstruction. Only then novel views can be rendered and edited, resulting in two separate rounds of 3D training: one for initial reconstruction, and another for learning from edited views. To address the first issue, we are exploring more efficient editing pipelines and view selection strategies (one of example should be our proposed Gaussian-based sampling) that reduce the number of novel views required without exhaustively computational overhead. For the second issue, we are investigating end-to-end training schemes where editing supervision can be integrated earlier in the reconstruction process, potentially reducing the need for two distinct learning stages.

## C  DETAILS OF OUTLIER REMOVAL AND 3D GAUSSIAN PROBABILISTIC VIEW SAMPLING

**Outlier Removal.**  To define a valid sampling region and avoid selecting novel viewpoints from semantically or geometrically irrelevant spaces (e.g., the back or bottom of the scene), we introduce an outlier removal strategy based on a cube bounding the main body of the 3D point cloud. Let the 3D structure's point cloud be denoted as $\mathcal{P} = \{\mathbf{p}_i\}_{i=1}^N \subset \mathbb{R}^3$, where each $\mathbf{p}_i = (x_i, y_i, z_i)^\top$ represents a 3D point and $N$ is the number of points. We first compute the mean (centroid) of the point cloud as:

$$\boldsymbol{\mu} = \frac{1}{N} \sum_{i=1}^N \mathbf{p}_i = (\mu_x, \mu_y, \mu_z)^\top. \tag{6}$$

Next, we calculate the maximum coordinate-wise deviation (i.e., the $L^\infty$ distance) of each point from the centroid:

$$d_i = \|\mathbf{p}_i - \boldsymbol{\mu}\|_\infty = \max\left(|x_i - \mu_x|, |y_i - \mu_y|, |z_i - \mu_z|\right). \tag{7}$$

To determine the size of the cube that includes approximately $90\%$ of the point cloud, we compute the 90th percentile of the distances $\{d_i\}_{i=1}^N$:

$$\delta_{90} = \text{Percentile}_{90}\left(\{d_i\}_{i=1}^N\right). \tag{8}$$

Using this threshold, we define the cube inclusion region $\mathcal{C}_{90} \subset \mathbb{R}^3$ centered at $\boldsymbol{\mu}$ as:

$$\mathcal{C}_{90} = \left\{\mathbf{p} \in \mathbb{R}^3 \,\middle|\, \|\mathbf{p} - \boldsymbol{\mu}\|_\infty \leq \delta_{90}\right\}. \tag{9}$$

During the sampling of novel viewpoints, any sampled viewpoint $\pi^*$ that falls inside this cube is regarded as lying within the exclusion space and is therefore discarded and resampled. This prevents oversampling in central, densely occupied regions of the scene, and encourages viewpoint exploration in surrounding, less-redundant areas. This approach can be effective when the point cloud is evenly distributed around the mean point along the x, y, and z axes. However, since point clouds can exhibit various distribution patterns, we further tuned the inclusion space by adding or subtracting offsets $\alpha = (\alpha_x, \alpha_x, \alpha_z)$ to the lengths of each side of the cube as $\mathcal{C}_{90}^\alpha = \left\{\mathbf{p} \in \mathbb{R}^3 \,\middle|\, \|\mathbf{p} - \boldsymbol{\mu} + \alpha\|_\infty \leq \delta_{90}\right\}$. Thus, $\mathcal{C}_{90}^\alpha$ is the final outlier removal space that we used. This inclusion/exclusion space is employed into 3D Gaussian Probabilistic View Sampling (3DG-PVS) as outlier removal to perform screen out ineffective sampled viewpoints (*e.g.*, back side, viewpoints too far away from target).

# D   DETAILS OF NOVEL VIEW SAMPLING PROCESS

We use an isotropic 3D Gaussian with a default standard deviation of $\sigma = 0.4$, which provides a stable balance between local density estimation and sufficient spatial smoothness. For balanced optimization between edited reference view images and edited novel view images, we use ratio of scaler $2 \leq \alpha \leq 3$ in Equation (4). The normalizing constant $Z$ in the inverse distribution of Equation (5) is approximated via Monte Carlo sampling using 50k uniformly drawn points inside a bounding box, which we found sufficient for stable estimation across all scenes. For constructing the sampling region, we compute a bounding box that contains 90% of the reference viewpoints by default, which ensures both robustness and minimal influence of outliers. Novel viewpoints are sampled with rejection sampling using the same bounding box, as it reliably covers the full distribution of reference viewpoints. The exclusion region is implemented as an axis-aligned box with optional offsets as $\alpha = \{\alpha_x, \alpha_y, \alpha_z\}$, bounded by $-0.5 < \alpha_i < 0.5$ ($i \in \{x, y, z\}$). These choices were empirically validated to provide stable sampling behavior across diverse scenes.

**3D Gaussian Probabilistic View Sampling.**   3DG-PVS is performed in a iterative approach by sampling each proper novel viewpoints based on previously sampled viewpoints and reference viewpoints.

---

**Algorithm 1:** 3D Gaussian Probabilistic View Sampling (3DG-PVS)

---

**Input   :**
- $\mathcal{X} = \{\pi_t\}_{t=1}^T$: Reference viewpoint set
- $T^*$: Number of novel viewpoints to sample
- Exclusion space for outlier removal (given above)
- Isotropic covariance $\Sigma_t = \sigma^2 I \in \mathbb{R}^{3 \times 3}$ (with $\sigma = 0.4$)

**Output :** $\mathcal{Y} = \{\pi_t^*\}_{t=1}^{T^*}$: Sampled novel viewpoints

**Initialize:** $\mathcal{Y} \leftarrow \emptyset$
**for** $k \leftarrow 1$ **to** $T^*$ **do**
 **if** $k = 1$ **then**
  Compute the mixture of Gaussians:

$$P(\pi_1^* \mid \mathcal{X}) = \frac{1}{T} \sum_{t=1}^T \mathcal{N}(\pi_1^* \mid \pi_t, \Sigma_t).$$

  Define the inverse probability:

$$\tilde{P}(\pi_1^* \mid \mathcal{X}) = \frac{1 - P(\pi_1^* \mid \mathcal{X})}{Z}, \quad Z = \int \big(1 - P(\pi_1^* \mid \mathcal{X})\big) \, d\pi_1^*.$$

  Sample $\pi_1^*$ from $\tilde{P}(\pi_1^* \mid \mathcal{X})$.
 **else**
  *# Update mixture by incorporating both*
  *# reference set and previously sampled novel views.*
  Compute updated mixture of Gaussians:

$$P(\pi_k^* \mid \mathcal{X}, \mathcal{Y}_{<k}) = \frac{1}{T + (k-1)} \Big( \sum_{t=1}^T \mathcal{N}(\pi_k^* \mid \pi_t, \Sigma_t) + \sum_{j=1}^{k-1} \mathcal{N}(\pi_k^* \mid \pi_j^*, \Sigma_j) \Big).$$

  Define inverse probability $\tilde{P}(\pi_k^* \mid \mathcal{X}, \mathcal{Y}_{<k})$ similarly and sample $\pi_k^*$ from it.
 *# Outlier removal check:*
 **if** $\pi_k^*$ *falls in excluded (outlier) region* **then**
  Discard $\pi_k^*$ and resample until a valid viewpoint is found.
 *# Add newly sampled viewpoint to set:*
 $\mathcal{Y} \leftarrow \mathcal{Y} \cup \{\pi_k^*\}$.
**return** $\mathcal{Y}$

---

# E  ADDITIONAL QUANTITATIVE RESULTS

Tables 4 and 5 are statistical analysis with mean and confidence interval using standard deviation with 30 runs of random seeds. Tables 6 and 7 present independent evaluations on two widely used 3D editing benchmarks. We observe that applying our NVE-Adaptor consistently improves quality, view-consistency, and textual alignment. Moreover, the improvements are comparable across both datasets, demonstrating the robustness of our approach to different data domains.

Table 8 analyzes performance improvements across different scene types. Specifically, we divide the IN2N and Mip-NeRF 360 datasets into four categories: indoor, outdoor, reflective surfaces, and cluttered layouts. To evaluate reflective surfaces in particular, we additionally utilize the materials category from the NeRF Synthetic dataset Mildenhall et al. (2020), which contains reflective objects. For each setting, we perform inference with 30 random seeds. While the magnitude of improvement varies slightly across scene types, our method consistently achieves performance gains in all categories.

Table 9 reports results on multiple-image editing with different prompts that explicitly request quality improvements. We find that most prompts contribute to a similar degree of enhancement, indicating that our framework is robust to prompt variation. Table 10 presents a sensitivity analysis of reference view sparsity. Experiments on 24 scenes are conducted by progressively reducing the number of reference views from 100 to 20. The step size of 20 was chosen as it is the smallest decrement that results in changes across all evaluation metrics. The NVE-Adaptor consistently provides performance gains down to 60 reference views. However, when the number of references drops below 40, the base 3D editing quality deteriorates sharply, limiting the effectiveness of our method. In extremely sparse settings (*e.g.*, 20 views), unseen-view renderings become too ambiguous or semantically unrecognizable, causing the diffusion editor to generate new content rather than refine existing structures. This mismatch leads to degraded supervision and reduced performance.

Table 11 presents an ablation study to examine whether using edited novel views may introduce texture hallucination and degrade quality. When the reference view and the edited novel view are provided as *independent* inputs to the diffusion model, we indeed observe quality degradation. However, when they are provided jointly through *concatenation*, the results consistently outperform using the reference view alone. This suggests that, when both inputs participate in the diffusion model's interaction mechanisms (e.g., attention), visually reliable information from the reference view helps guide and stabilize the edited novel view, leading to improved quality. Table 12 shows a sensitivity analysis of the 3DG-PVS with respect to the Gaussian parameter $\sigma$. Across a wide range of $\sigma$ values, the sampling strategy remains stable and produces robust performance.

Table 13 provides a detailed analysis of time and memory consumption for each baseline model before and after applying our method. For training, we break down the wall-clock time into three components: diffusion-based editing, rendering, and 3D optimization. For inference, we additionally report the novel-view rendering speed (fps) for typical scenes. We also include GPU memory statistics, such as total GPU hours and peak memory usage for the entire training pipeline, as well as the GPU hours and memory specifically attributable to the diffusion editing stage. Overall, this analysis demonstrates that our method introduces only minimal computational and memory overhead, showing that it is both resource-efficient and time-efficient.

Table 14 provides a concise comparison of our 3D Gaussian Probabilistic View Sampling (3DG-PVS) with two vanilla alternatives: random sampling and uniform sampling. We summarize the four essential criteria for effective novel-view selection—unexplored-region coverage, sparsity, efficiency, and outlier handling—and evaluate how each strategy satisfies them. As shown in the table, 3DG-PVS is the only method that simultaneously meets all four criteria: it targets uncovered regions via inverse-probability sampling, maintains sparsity through iterative updates, removes implausible viewpoints, and achieves high coverage with a small number of views. In contrast, random sampling is unstable and redundant, while uniform sampling is computationally inefficient and lacks scene awareness. This analysis highlights why 3DG-PVS serves as a principled and effective sampling strategy for 3D editing. Table 15 provides a fair, equal-time comparison between our method and the baselines. The additional views used by our approach are not extra ground-truth images but unseen views rendered from the existing 3D structure. Even when given the same training time as each baseline, our method consistently achieves superior performance, confirming that the improvement comes from effective view sampling rather than additional data or computation.

Table 4: Quantitative evaluations of NVE-Adaptor with 3D editing systems on IN2N and Mip-NeRF 360 datasets. $CLIP_I$ is image-to-image clip score for consistency. It is reported in a format of (original test set / unseen view set). All scores are 'mean $+ \pm 1.96 \times$ (standard deviation / $\sqrt{n}$)' with $n = 30$, following 95% confidence interval.

| Method | Consistency | | Fidelity | |
|---|---|---|---|---|
| | FID ↓ | $CLIP_I$ ↑ | LPIPS ↓ | SSIM ↑ |
| IN2N Haque et al. (2023) | 254±5 / 306±11 | 0.905±0.002 / 0.871±0.009 | 0.374±0.005 / 0.449±0.008 | 0.627±0.007 / 0.562±0.009 |
| IN2N + NVE-Adaptor | 231±6 / 260±7 | 0.913±0.003 / 0.902±0.010 | 0.332±0.006 / 0.375±0.009 | 0.662±0.008 / 0.618±0.010 |
| GCtrl Wu et al. (2024) | 246±11 / 295±15 | 0.907±0.003 / 0.871±0.007 | 0.357±0.007 / 0.425±0.013 | 0.636±0.012 / 0.581±0.014 |
| GCtrl + NVE-Adaptor | 215±7 / 247±9 | 0.914±0.004 / 0.902±0.006 | 0.311±0.007 / 0.361±0.015 | 0.675±0.011 / 0.622±0.014 |
| VcEdit Wang et al. (2024) | 231±9 / 281±17 | 0.914±0.004 / 0.879±0.004 | 0.339±0.005 / 0.419±0.011 | 0.654±0.009 / 0.608±0.013 |
| VcEdit + NVE-Adaptor | 209±7 / 242±11 | 0.924±0.005 / 0.908±0.006 | 0.301±0.007 / 0.347±0.014 | 0.693±0.010 / 0.649±0.015 |
| DGE Chen et al. (2024) | 224±6 / 270±8 | 0.920±0.003 / 0.887±0.004 | 0.328±0.009 / 0.401±0.016 | 0.672±0.011 / 0.613±0.018 |
| DGE + NVE-Adaptor | 200±5 / 234±8 | 0.929±0.002 / 0.916±0.005 | 0.295±0.011 / 0.336±0.017 | 0.705±0.014 / 0.657±0.019 |

Table 5: Quantitative evaluations of NVE-Adaptor with 3D editing systems on IN2N and Mip-NeRF 360 datasets. $CLIP_T$ is text-to-image clip for textual alignment and $CLIP_D$ is directional clip score. It is reported in a format of (original test set / unseen view set). All scores are 'mean $+ \pm 1.96 \times$ (standard deviation / $\sqrt{n}$)' with $n = 30$, following 95% confidence interval.

| Method | Textual Alignment | | Human |
|---|---|---|---|
| | $CLIP_T$ ↑ | $CLIP_D$ ↑ | |
| IN2N Haque et al. (2023) | 0.199±0.001 / 0.184±0.002 | 0.048±0.001 / 0.037±0.001 | 0.22 |
| IN2N + NVE-Adaptor | 0.207±0.002 / 0.195±0.002 | 0.060±0.001 / 0.049±0.001 | 0.78 |
| GCtrl Wu et al. (2024) | 0.202±0.001 / 0.184±0.002 | 0.056±0.002 / 0.039±0.002 | 0.25 |
| GCtrl + NVE-Adaptor | 0.212±0.001 / 0.196±0.003 | 0.064±0.002 / 0.054±0.003 | 0.75 |
| VcEdit Wang et al. (2024) | 0.210±0.002 / 0.185±0.003 | 0.061±0.001 / 0.047±0.001 | 0.34 |
| VcEdit + NVE-Adaptor | 0.217±0.001 / 0.205±0.002 | 0.067±0.002 / 0.061±0.002 | 0.66 |
| DGE Chen et al. (2024) | 0.207±0.001 / 0.190±0.002 | 0.062±0.001 / 0.046±0.002 | 0.33 |
| DGE + NVE-Adaptor | 0.215±0.002 / 0.207±0.002 | 0.068±0.001 / 0.062±0.002 | 0.67 |

Table 6: Quantitative evaluations of NVE-Adaptor with recent 3D editing systems on Mip-NeRF 360 dataset. $CLIP_T$ is text-to-image clip for textual alignment, $CLIP_D$ is directional clip score, and $CLIP_I$ is image-to-image clip score for consistency. All scores are 'mean $+ \pm 1.96 \times$ (standard deviation / $\sqrt{n}$)' with $n = 30$, following 95% confidence interval.

| Method | Consistency | | Fidelity | | Textual Alignment | | Human |
|---|---|---|---|---|---|---|---|
| | FID ↓ | $CLIP_I$ ↑ | LPIPS ↓ | SSIM ↑ | $CLIP_T$ ↑ | $CLIP_D$ ↑ | |
| IN2N Haque et al. (2023) | 275±5 | 0.890±0.001 | 0.401±0.005 | 0.607±0.006 | 0.191±0.002 | 0.044±0.001 | 0.23 |
| IN2N + NVE-Adaptor | 240±7 | 0.909±0.002 | 0.351±0.006 | 0.648±0.005 | 0.204±0.001 | 0.054±0.001 | 0.77 |
| GCtrl Wu et al. (2024) | 263±14 | 0.892±0.002 | 0.380±0.012 | 0.617±0.007 | 0.195±0.001 | 0.050±0.002 | 0.31 |
| GCtrl + NVE-Adaptor | 228±12 | 0.911±0.003 | 0.330±0.007 | 0.654±0.012 | 0.206±0.002 | 0.062±0.002 | 0.69 |
| VcEdit Wang et al. (2024) | 251±10 | 0.899±0.001 | 0.363±0.011 | 0.637±0.008 | 0.201±0.001 | 0.057±0.001 | 0.35 |
| VcEdit + NVE-Adaptor | 218±11 | 0.916±0.001 | 0.321±0.010 | 0.678±0.007 | 0.214±0.001 | 0.063±0.002 | 0.65 |
| DGE Chen et al. (2024) | 242±8 | 0.902±0.003 | 0.361±0.009 | 0.648±0.010 | 0.201±0.001 | 0.056±0.001 | 0.37 |
| DGE + NVE-Adaptor | 215±9 | 0.921±0.002 | 0.319±0.011 | 0.685±0.011 | 0.210±0.002 | 0.063±0.001 | 0.63 |

Table 7: Quantitative evaluations of NVE-Adaptor with recent 3D editing systems on multiple scenes from IN2N dataset. $CLIP_T$ is text-to-image clip for textual alignment, $CLIP_D$ is directional clip score, and $CLIP_I$ is image-to-image clip score for consistency. All scores are 'mean $+ \pm 1.96 \times$ (standard deviation / $\sqrt{n}$)' with $n = 30$, following 95% confidence interval.

| Method | Consistency | | Fidelity | | Textual Alignment | | Human |
|---|---|---|---|---|---|---|---|
| | FID ↓ | $CLIP_I$ ↑ | LPIPS ↓ | SSIM ↑ | $CLIP_T$ ↑ | $CLIP_D$ ↑ | |
| IN2N Haque et al. (2023) | 248±5 | 0.910±0.002 | 0.367±0.004 | 0.630±0.004 | 0.201±0.002 | 0.051±0.001 | 0.21 |
| IN2N + NVE-Adaptor | 227±4 | 0.916±0.001 | 0.321±0.007 | 0.674±0.005 | 0.211±0.001 | 0.062±0.001 | 0.79 |
| GCtrl Wu et al. (2024) | 238±8 | 0.912±0.001 | 0.347±0.008 | 0.646±0.006 | 0.204±0.002 | 0.059±0.002 | 0.22 |
| GCtrl + NVE-Adaptor | 213±6 | 0.918±0.002 | 0.303±0.011 | 0.681±0.009 | 0.212±0.001 | 0.066±0.003 | 0.78 |
| VcEdit Wang et al. (2024) | 225±7 | 0.920±0.003 | 0.328±0.009 | 0.662±0.011 | 0.213±0.002 | 0.065±0.001 | 0.33 |
| VcEdit + NVE-Adaptor | 204±10 | 0.926±0.001 | 0.291±0.009 | 0.702±0.008 | 0.220±0.002 | 0.070±0.002 | 0.67 |
| DGE Chen et al. (2024) | 218±5 | 0.926±0.001 | 0.322±0.010 | 0.678±0.008 | 0.211±0.001 | 0.064±0.001 | 0.31 |
| DGE + NVE-Adaptor | 197±4 | 0.932±0.002 | 0.286±0.008 | 0.714±0.007 | 0.217±0.002 | 0.071±0.002 | 0.69 |

Table 8: Quantitative results across diverse scene types including indoor, outdoor, cluttered layouts, and reflective surfaces. The baseline scores are the average performances of Instruct-NeRF2NeRF, GaussCtrl, VcEdit, and DGE. "NVE" refers to NVE-Adaptor. $CLIP_T$ is text-to-image clip for textual alignment, $CLIP_D$ is directional clip score, and $CLIP_I$ is image-to-image clip score for consistency. (30 runs with random seed). All scores are 'mean $+ \pm 1.96 \times$ (standard deviation / $\sqrt{n}$)' with $n = 30$, following 95% confidence interval.

| Method | Consistency | | Fidelity | | Textual Alignment | |
|---|---|---|---|---|---|---|
| | FID $\downarrow$ | $CLIP_I \uparrow$ | LPIPS $\downarrow$ | SSIM $\uparrow$ | $CLIP_T \uparrow$ | $CLIP_D \uparrow$ |
| Indoor (Baseline) | 231±5 | 0.916±0.003 | 0.306±0.003 | 0.706±0.002 | 0.206±0.001 | 0.058±0.001 |
| Indoor (Baseline + NVE) | 205±4 | 0.925±0.001 | 0.291±0.004 | 0.724±0.004 | 0.211±0.001 | 0.070±0.002 |
| Outdoor (Baseline) | 234±9 | 0.911±0.003 | 0.299±0.005 | 0.701±0.004 | 0.203±0.002 | 0.060±0.002 |
| Outdoor (Baseline + NVE) | 217±7 | 0.920±0.003 | 0.286±0.006 | 0.712±0.006 | 0.209±0.002 | 0.067±0.003 |
| reflective surfaces (Baseline) | 281±5 | 0.896±0.001 | 0.352±0.004 | 0.658±0.004 | 0.229±0.002 | 0.055±0.002 |
| reflective surfaces (Baseline + NVE) | 246±6 | 0.911±0.002 | 0.334±0.004 | 0.691±0.005 | 0.235±0.001 | 0.064±0.002 |
| cluttered layouts (Baseline) | 279±12 | 0.889±0.004 | 0.356±0.008 | 0.661±0.008 | 0.197±0.002 | 0.051±0.003 |
| cluttered layouts (Baseline + NVE) | 248±11 | 0.919±0.005 | 0.331±0.011 | 0.689±0.010 | 0.204±0.003 | 0.063±0.003 |

Table 9: Quantitative results according to different prompts for diffusion editing. It is reported in a format of (original test set / unseen view set) of Table 1 of main paper with average of the four baselines (Instruct-NeRF2NeRF, GaussCtrl, VcEdit, and DGE). $CLIP_T$ is text-to-image clip for textual alignment, $CLIP_D$ is directional clip score, and $CLIP_I$ is image-to-image clip score for consistency.

| Prompt | Consistency | | Fidelity | | Textual Alignment | |
|---|---|---|---|---|---|---|
| | FID $\downarrow$ | $CLIP_I \uparrow$ | LPIPS $\downarrow$ | SSIM $\uparrow$ | $CLIP_T \uparrow$ | $CLIP_D \uparrow$ |
| No use | 228/276 | 0.911/0.882 | 0.338/0.407 | 0.651/0.607 | 0.209/0.190 | 0.060/0.047 |
| "high resolution" | 212/237 | 0.918/0.904 | 0.303/0.351 | 0.690/0.665 | 0.211/0.200 | 0.064/0.058 |
| "photo-realistic" | 211/233 | 0.919/0.906 | 0.301/0.346 | 0.693/0.669 | 0.212/0.204 | 0.066/0.060 |
| "8k resolution" | 209/233 | 0.921/0.906 | 0.299/0.347 | 0.691/0.666 | 0.212/0.201 | 0.068/0.061 |
| "high quality" | 210/234 | 0.920/0.907 | 0.300/0.347 | 0.692/0.668 | 0.213/0.203 | 0.067/0.063 |

Table 10: Sensitivity analysis according to the number of input reference viewpoints (N). (Reported as average scores of original test set and unseen view set in Table 1 of main paper). NVE denotes to NVE-Adaptor and Base denotes average scores of baselines (Instruct-NeRF2NeRF, GaussCtrl, VcEdit, DGE). $CLIP_T$ is text-to-image clip for textual alignment, $CLIP_D$ is directional clip score, and $CLIP_I$ is image-to-image clip score for consistency.

| Method | Consistency | | Fidelity | | Textual Alignment | |
|---|---|---|---|---|---|---|
| | FID $\downarrow$ | $CLIP_I \uparrow$ | LPIPS $\downarrow$ | SSIM $\uparrow$ | $CLIP_T \uparrow$ | $CLIP_D \uparrow$ |
| Base (N=100) | 246 | 0.908 | 0.356 | 0.659 | 0.205 | 0.054 |
| Base + NVE (N=100) | 213 | 0.928 | 0.317 | 0.689 | 0.211 | 0.066 |
| Base (N=80) | 249 | 0.906 | 0.358 | 0.656 | 0.204 | 0.052 |
| Base + NVE (N=80) | 211 | 0.929 | 0.317 | 0.691 | 0.210 | 0.065 |
| Base (N=60) | 254 | 0.902 | 0.366 | 0.651 | 0.200 | 0.050 |
| Base + NVE (N=60) | 211 | 0.927 | 0.319 | 0.689 | 0.211 | 0.066 |
| Base (N=40) | 260 | 0.898 | 0.371 | 0.646 | 0.198 | 0.047 |
| Base + NVE (N=40) | 232 | 0.906 | 0.358 | 0.659 | 0.204 | 0.053 |
| Base (N=20) | 293 | 0.875 | 0.376 | 0.619 | 0.184 | 0.042 |
| Base + NVE (N=20) | 295 | 0.873 | 0.376 | 0.616 | 0.184 | 0.040 |

Table 11: 3D reconstruction results with variations of NVE-Adaptor in terms of input image setting (i.e., reference view images, edited novel view images) used in multi-view consistent diffusion models. We report the average scores of four baseline settings (i.e., IN2N, GCtrl, VcEdit, DGE). $CLIP_I$ is image-to-image clip score for consistency.

| Prompt | Consistency | Fidelity | |
|---|---|---|---|
| | $CLIP_I \uparrow$ | LPIPS $\downarrow$ | SSIM $\uparrow$ |
| (A) Reference only | 0.905 | 0.291 | 0.762 |
| (B) Independent processing reference and edited novel view | 0.886 | 0.298 | 0.756 |
| (C) Concatenated processing reference and edited novel view (1:1) | 0.915 | 0.258 | 0.783 |
| (D) Concatenated processing reference and edited novel view (2:1) | 0.909 | 0.278 | 0.778 |
| (E) Concatenated processing reference and edited novel view (1:2) | 0.928 | 0.237 | 0.793 |
| (F) Concatenated processing reference and edited novel view (1:3) | 0.938 | 0.226 | 0.806 |
| (G) Concatenated processing reference and edited novel view (1:4) | 0.935 | 0.229 | 0.801 |

Table 12: Sensitivity analysis according to variants of $\sigma$ with average scores of baselines. It is reported in a format of (original test set / unseen view set) of Table 1 of main paper with average of the four baselines (Instruct-NeRF2NeRF, GaussCtrl, VcEdit, and DGE) with NVE-Adaptor. $CLIP_T$ is text-to-image clip for textual alignment, $CLIP_D$ is directional clip score, and $CLIP_I$ is image-to-image clip score for consistency.

| $\sigma$ | Consistency | | Fidelity | | Textual Alignment | |
|---|---|---|---|---|---|---|
| | FID $\downarrow$ | $CLIP_I \uparrow$ | LPIPS $\downarrow$ | SSIM $\uparrow$ | $CLIP_T \uparrow$ | $CLIP_D \uparrow$ |
| 0.2 | 222 / 251 | 0.915 / 0.904 | 0.310 / 0.364 | 0.680 / 0.630 | 0.211 / 0.200 | 0.061 / 0.051 |
| 0.3 | 215 / 246 | 0.918 / 0.906 | 0.308 / 0.358 | 0.684 / 0.634 | 0.212 / 0.201 | 0.064 / 0.055 |
| 0.4 | 215 / 245 | 0.921 / 0.908 | 0.307 / 0.356 | 0.686 / 0.636 | 0.214 / 0.202 | 0.067 / 0.057 |
| 0.5 | 217 / 247 | 0.920 / 0.908 | 0.306 / 0.356 | 0.687 / 0.636 | 0.214 / 0.202 | 0.066 / 0.056 |
| 0.6 | 219 / 249 | 0.918 / 0.906 | 0.309 / 0.358 | 0.685 / 0.634 | 0.213 / 0.202 | 0.065 / 0.054 |
| 0.7 | 221 / 250 | 0.917 / 0.905 | 0.311 / 0.363 | 0.682 / 0.631 | 0.213 / 0.202 | 0.063 / 0.053 |
| 0.8 | 227 / 254 | 0.913 / 0.902 | 0.315 / 0.367 | 0.678 / 0.628 | 0.210 / 0.198 | 0.059 / 0.049 |

Table 13: Training time in terms of three components, inference speed, and memory consumption .

| Method | Training time 1: diffusion editing | Training time 2: rendering | Training time 3: 3D optimization | Inference speed | Total GPU hours | Peak memory |
|---|---|---|---|---|---|---|
| IN2N Haque et al. (2023) | 4 min | 20 min | 32 min | 0.09 fps | 2 GPU hours | 7.9 GB |
| IN2N + NVE-Adaptor | 5 min | 25 min | 38 min | 0.09 fps | 2 GPU hours | 8.0 GB |
| GCtrl Wu et al. (2024) | 5 min | < 1 min | 17 min | 91 fps | < 1 GPU hours | 13.4 GB |
| GCtrl + NVE-Adaptor | 6 min | < 1 min | 21 min | 89 fps | < 1 GPU hours | 13.6 GB |
| VcEdit Wang et al. (2024) | 5 min | < 1 min | 15 min | 94 fps | < 1 GPU hours | 13.7 GB |
| VcEdit + NVE-Adaptor | 6 min | < 1 min | 19 min | 91 fps | < 1 GPU hours | 13.8 GB |
| DGE Chen et al. (2024) | 3 min | < 1 min | 11 min | 92 fps | < 1 GPU hours | 15.4 GB |
| DGE + NVE-Adaptor | 4 min | < 1 min | 13 min | 90 fps | < 1 GPU hours | 15.7 GB |

Table 14: Quantitative comparison with random and uniform sampling strategies. We report the average scores across four baselines (IN2N, GCtrl, VcEdit, and DGE). Note that for uniform sampling, we applied our outlier removal to ensure computational tractability; otherwise, the novel view count would be intractable. For random sampling and our sampling, the number of views is fixed at 30.

| Method | Unexplored region | Novel view sparsity | Efficiency | Outlier handling | Seen view set | | Unseen view set | | Training Time |
|---|---|---|---|---|---|---|---|---|---|
| | | | | | FID $\downarrow$ | SSIM $\uparrow$ | FID $\downarrow$ | SSIM $\uparrow$ | |
| Random | X | X | X | X | 236 | 0.654 | 286 | 0.598 | 36 min |
| Uniform | O | O | X | X | 217 | 0.688 | 264 | 0.610 | 124 min |
| 3DG-PVS | O | O | O | O | 212 | 0.693 | 230 | 0.651 | 35 min |

Table 15: Fair comparison analysis with same training time with and without NVE-Adaptor. It is reported in a format of (original test set / unseen view set) of Table 1 of main paper with average of the four baselines (Instruct-NeRF2NeRF, GaussCtrl, VcEdit, and DGE) with NVE-Adaptor. $\text{CLIP}_T$ is text-to-image clip for textual alignment, $\text{CLIP}_D$ is directional clip score, and $\text{CLIP}_I$ is image-to-image clip score for consistency.

| Mthod | Time (min) | Consistency | | Fidelity | | Textual Alignment | |
|---|---|---|---|---|---|---|---|
| | | FID ↓ | $\text{CLIP}_I$ ↑ | LPIPS ↓ | SSIM ↑ | $\text{CLIP}_T$ ↑ | $\text{CLIP}_D$ ↑ |
| IN2N Haque et al. (2023) | 56 min | 256 / 307 | 0.904 / 0.870 | 0.377 / 0.442 | 0.623 / 0.576 | 0.198 / 0.183 | 0.048 / 0.035 |
| IN2N + NVE | 56 min | 234 / 265 | 0.913 / 0.900 | 0.334 / 0.389 | 0.664 / 0.615 | 0.207 / 0.193 | 0.058 / 0.046 |
| GCtrl Wu et al. (2024) | 23 min | 245 / 294 | 0.906 / 0.872 | 0.356 / 0.423 | 0.637 / 0.583 | 0.210 / 0.186 | 0.056 / 0.039 |
| GCtrl + NVE | 23 min | 220 / 253 | 0.914 / 0.901 | 0.313 / 0.365 | 0.670 / 0.620 | 0.209 / 0.197 | 0.064 / 0.054 |
| VcEdit Wang et al. (2024) | 21 min | 232 / 284 | 0.913 / 0.878 | 0.338 / 0.417 | 0.654 / 0.606 | 0.209 / 0.185 | 0.062 / 0.047 |
| VcEdit + NVE | 21 min | 213 / 242 | 0.921 / 0.909 | 0.302 / 0.348 | 0.690 / 0.644 | 0.218 / 0.205 | 0.067 / 0.059 |
| DGE Chen et al. (2024) | 15 min | 223 / 271 | 0.921 / 0.886 | 0.329 / 0.399 | 0.670 / 0.614 | 0.208 / 0.190 | 0.062 / 0.047 |
| DGE + NVE | 15 min | 203 / 235 | 0.928 / 0.915 | 0.296 / 0.339 | 0.702 / 0.651 | 0.213 / 0.205 | 0.067 / 0.060 |

# F NOVEL VIEW IMAGES AND EDITED NOVEL VIEW IMAGES

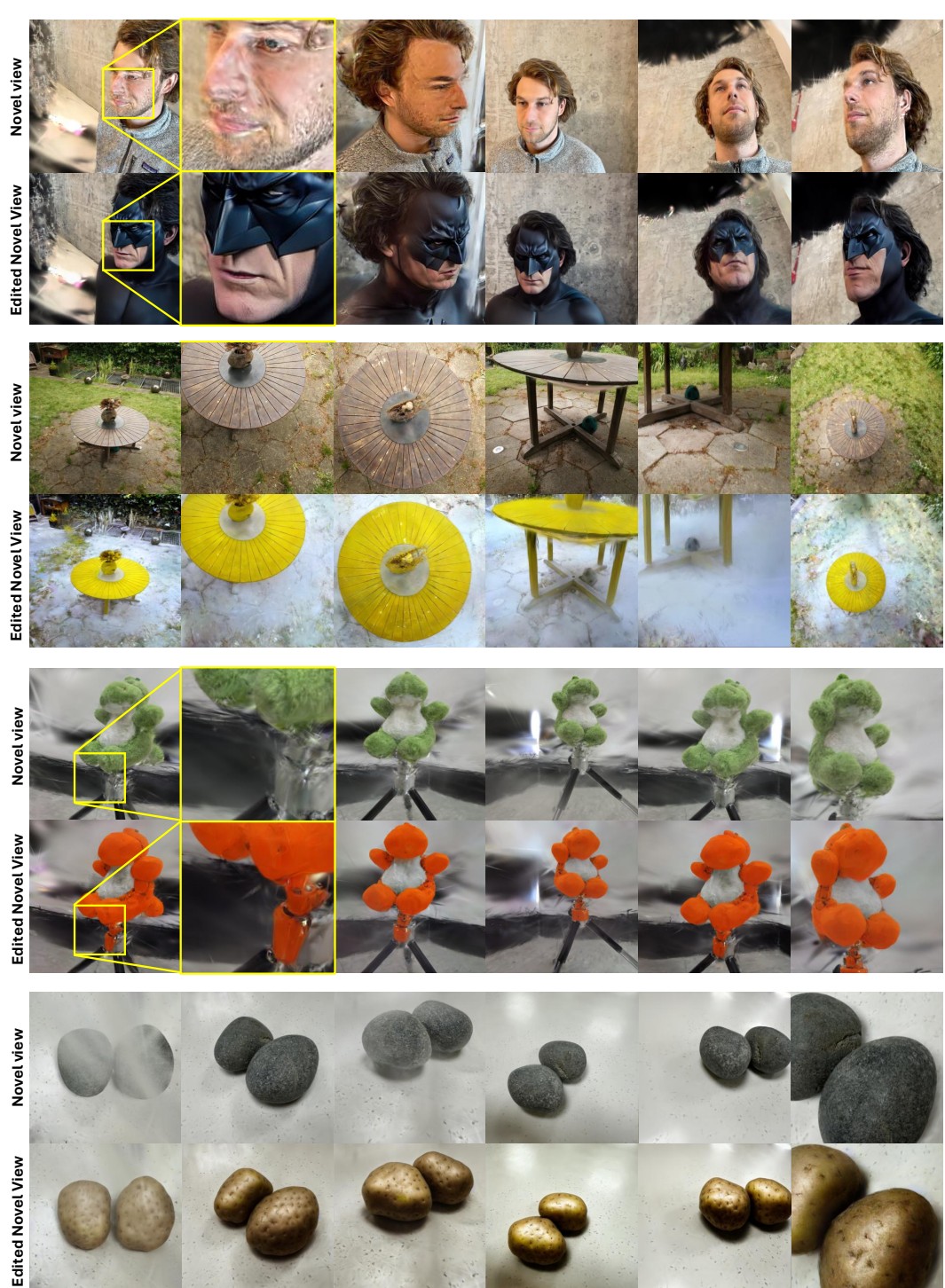

Figure 9: Novel view images using our NVE-Adaptor, along with the corresponding edited novel view images used as supervision for 3D editing. Editing improves the quality of low-resolution images rendered from novel viewpoints.

## G  ADDITIONAL QUALITATIVE RESULTS

Figure 10 presents results where various models integrated with NVE-Adaptor, are applied to the same subject. It also provides examples of different editing operations such as object overlay and style transfer on the same subject.

The top of Figure 11 shows the editing results from various viewpoints, as presented in the Introduction section of the main paper. The bottom of Figure 11 displays the results of applying different editing operations to the same subject, as shown in Figure 5 of the main paper.

Figure 12 shows the results of 3D editing on real-world images, comparing the outcomes before and after applying the NVE-Adaptor.

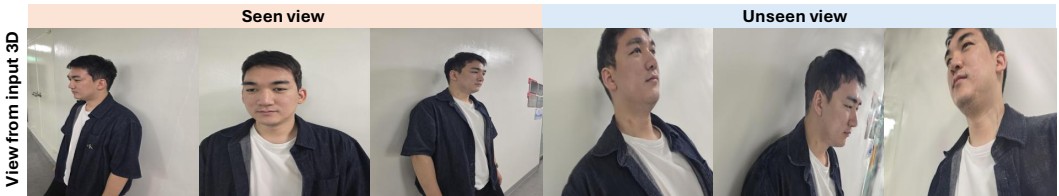

"Turn him into old man"

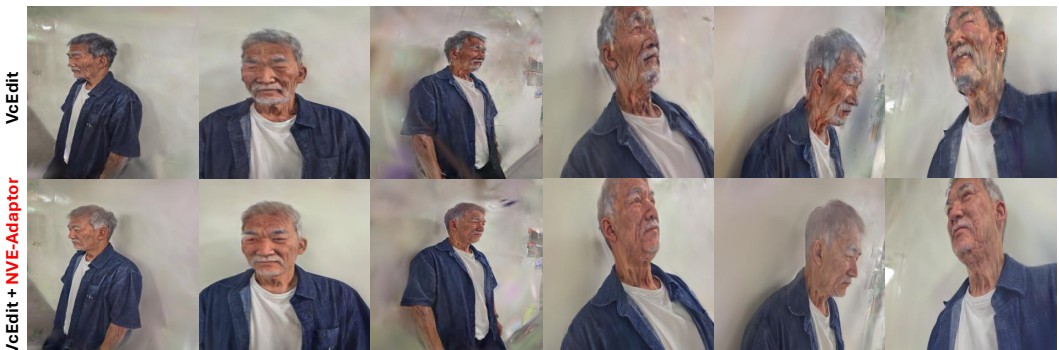

"Van gogh style image"

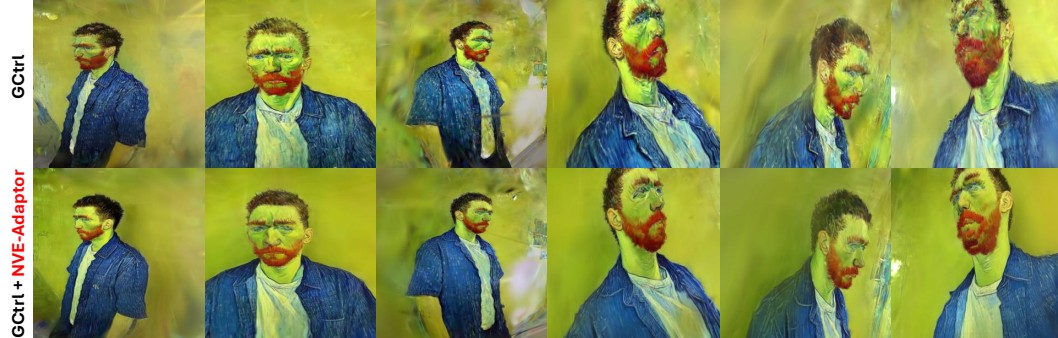

Figure 10: Results of different 3D editing applied to a single object. The first (top) editing shows results for object overlay, while the second (bottom) presents results for style transfer. Our model maintains consistent quality even under unseen novel views. (GCtrl: GaussCtrl)

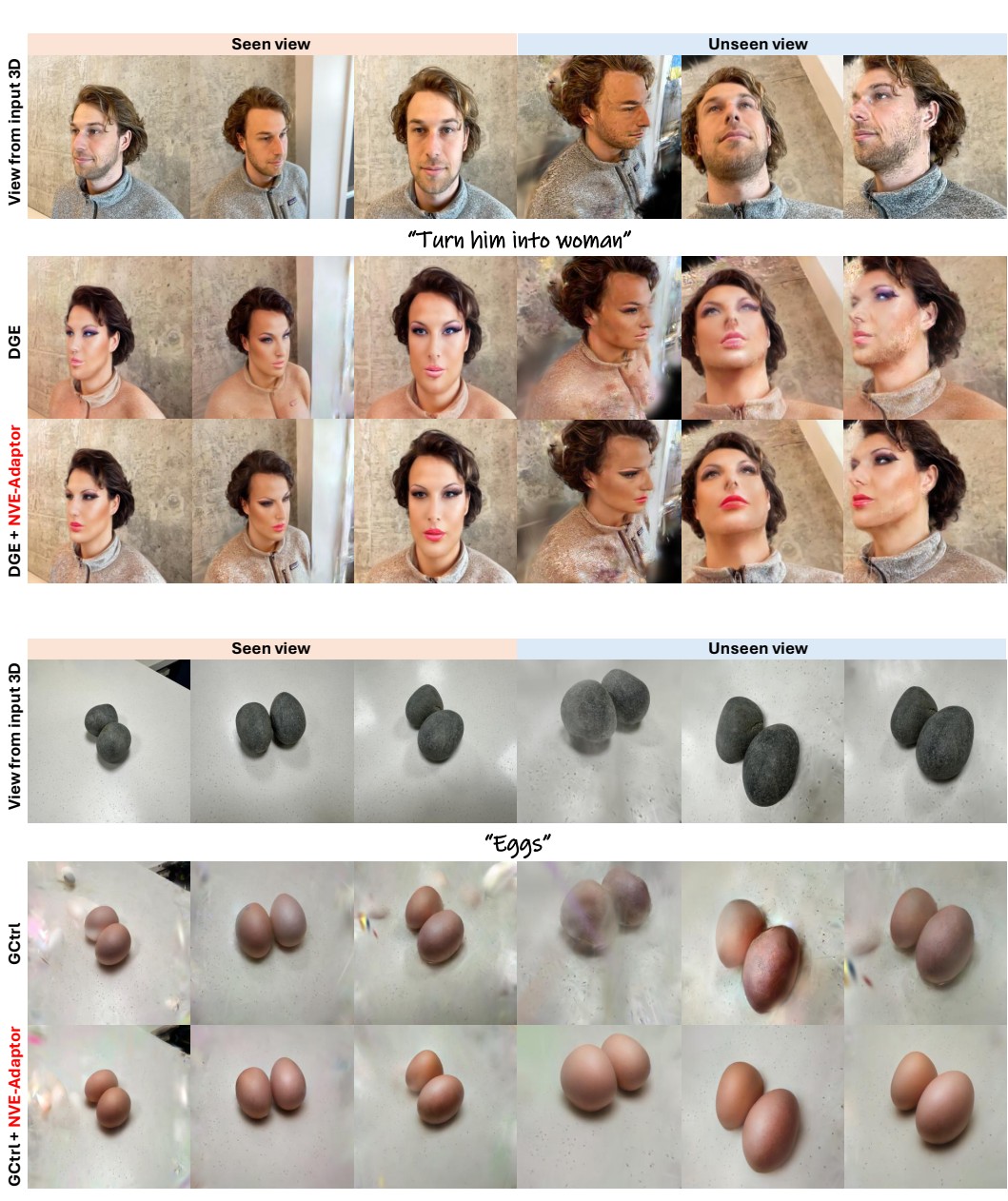

Figure 11: Additional editing results on samples in Figure 5 of the main paper. (GCtrl: GaussCtrl).

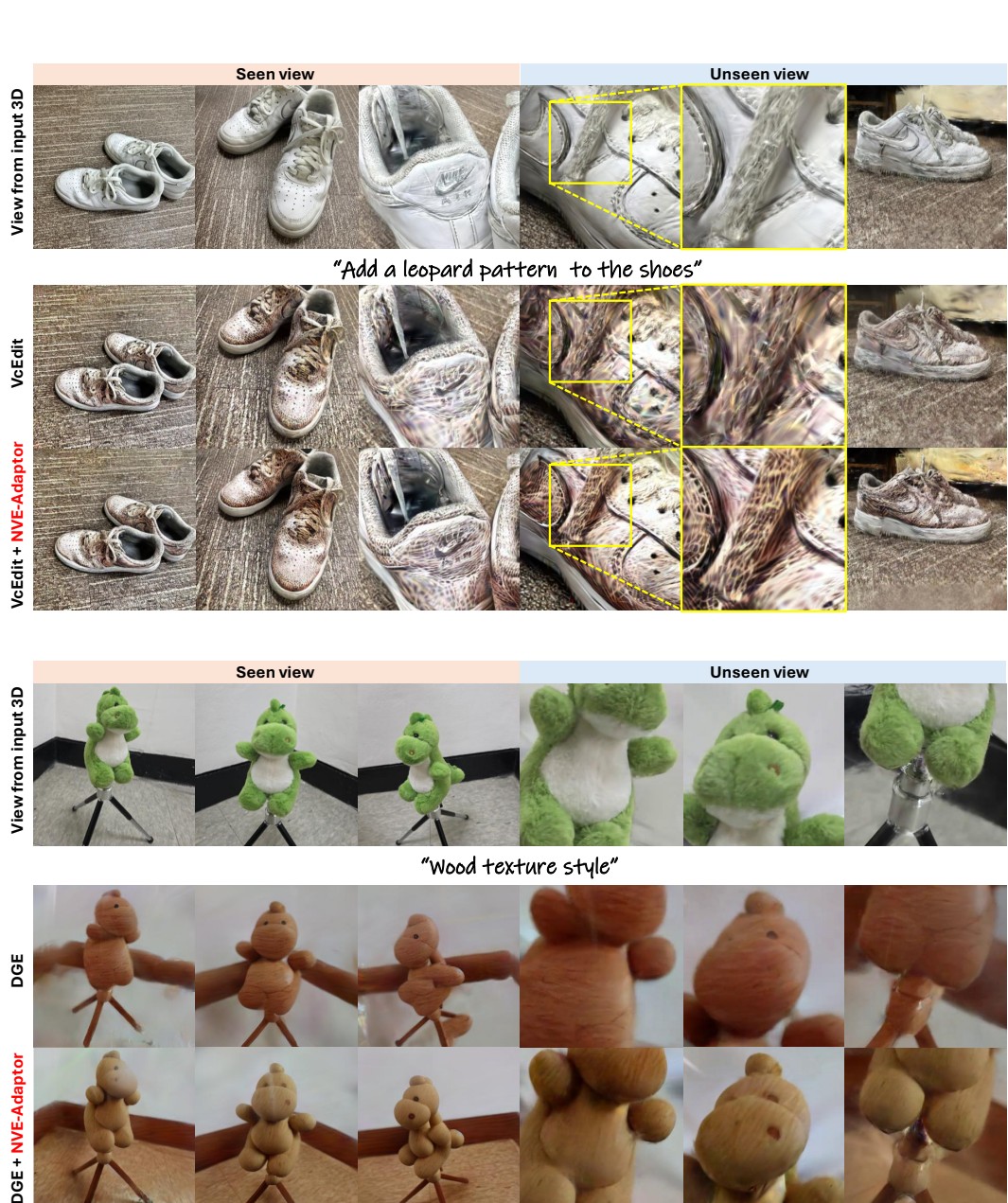

Figure 12: Examples of editing results on samples used in the novel view test set.

