# OpenReview forum: "NVE-Adaptor: Novel View Editing Adaptor for Unseen View Consistent 3D Editing"
_ICLR.cc/2026/Conference — Submitted to ICLR 2026_

### Official Review · Reviewer_CjWi · 2025-10-26

**Soundness:** 2
**Presentation:** 2
**Contribution:** 2
**Rating:** 2
**Confidence:** 3

**Summary:**

This paper proposes a novel view editing adaptor (NVE-Adaptor) for the 3D editing task. The main idea is to explore more novel views in 3D space and use Diffusion model to edit those novel views. Then the novel views and original reference views are combined to optimise the 3D scenes such as NeRF or 3D Gaussian Splatting.

The main contritution is the 3D Gaussian Probabilistic View Sampling (3DG-PVS) module, which explore novel views in a more effective way for 3D editing.

Experiments are conducted on Instruct-NeRF2NeRF (IN2N) and Mip-NeRF 360 datasets. The proposed method NVE-Adaptor is added to several 3D editing methods, and improves the performance w.r.t. several metrics.

**Strengths:**

## Strengths
- Illustration of vulnerability of unseen view rendering results of current 3D editing systems. This paper explores this phenomenen from two perspectives: angle ($\phi>25$) and distance ($r<0.5$). The visualization and continous set of angle/distance analysis show the severe degration of editing effect on novel views far from reference views.
- The Novel View Editing-Adaptor (NVE-Adaptor), at its core, it proposes a 3D GAUSSIAN PROBABILISTIC VIEW SAMPLING strategy for more effective sampling of new views for 3D editing.
- After sampling more views and utilizing them in the Consistent Diffusion Model, the proposed method can be combined with various method, and improve the performance.

**Weaknesses:**

## Weaknesses
- Even the vulnerability of unseen view is comprehensively explored and visualized, this behaviour is a well-know common sense [R1-R3]. Simply mentioning this phenomenen is not a strong contribution.
- Overall, the proposed method is very simple, selecting more views for optimising the 3D scene (shown in Figure 2). Trivially modifying Eq. 4 to include novel views. And the newly added novel views go the same procedure as previous ones: Diffusion Model editing. Only difference is adding a new prompt "high quality", which is qutie simple.
- 3D Gaussian Probabilistic View Sampling (3DG-PVS) can be a minor contribution, which samples more views by iteratively selecting the inverse probability. But no other comparison strategies except for the default 'Regular' strategy is compared.
- The performance is improved after combined with several methods. Howevere, this requries much more views to edit the 3D scene, which is an unfair comparison. In general, more views for optimisation should have better performance. Even the Regular strategy might have better performance when combined with the baseline methods.
- Are there other novel view selection/sampling methods? This should be surveyed and discussed in the related work and in the experiments.
- How is Human performance evaluated? Why the number differs so much e.g., 0.22 VS. 0.78 while the other metrics distinctions are not that significant?
- In Table 2, T*=40 is better than T*=30, wrongly highlighted the number.

[R1] Remondino, F., Karami, A., Yan, Z., Mazzacca, G., Rigon, S., & Qin, R. (2023). A critical analysis of NeRF-based 3D reconstruction. Remote Sensing, 15(14), 3585.
[R2] Zhang, J., Zhang, Y., Fu, H., Zhou, X., Cai, B., Huang, J., ... & Tang, X. (2022). Ray priors through reprojection: Improving neural radiance fields for novel view extrapolation. In Proceedings of the IEEE/CVF Conference on Computer Vision and Pattern Recognition (pp. 18376-18386).
[R3] Hull, M., Yang, H., Mehta, P., Phute, M., Cho, A., Wang, H., ... & Chau, P. (2025). 3D Gaussian Splat Vulnerabilities. arXiv preprint arXiv:2506.00280.

**Questions:**

Are there other novel view selection/sampling methods? This should be surveyed and discussed in the related work and in the experiments.

How is Human performance evaluated? Why the number differs so much e.g., 0.22 VS. 0.78 while the other metrics distinctions are not that significant?

---

> ### Author Response · Authors · 2025-12-02
>
> [**Q1**] (1) The vulnerability of unseen views is a well-known common sense [R1-R3]. (2) Simply mentioning this phenomenon is not a strong contribution.
>
> [**A1**] (1) The works [R1–R3] are all 3D reconstruction, not 3D editing. To the best of our knowledge, no prior work has studied novel-view quality degradation or improvement in the context of 3D editing. Existing editing methods evaluate edits only on the reference views and do not examine consistency under unseen viewpoints. Our analysis therefore fills an unexplored gap in the editing literature. (2) Moreover, we do not merely point out the phenomenon. We propose a solution specifically tailored for 3D editing. Unlike reconstruction, 3D editing already has access to a reconstructed 3D structure and a pretrained diffusion model, making our method easy to integrate without additional reconstruction stages. Applying the same idea to reconstruction would require a two-stage pipeline (reconstruct → enhance → reconstruct), which is far less efficient. Finally, because latent diffusion models excel at semantic transformations, our method yields larger improvements in editing than in reconstruction, as also reflected in Figures 5 and 8.
>
> [**Q2**] (1) Overall, the proposed method is very simple. (2) Are there other novel view selection methods?
>
> [**A2**] (1) Our method is intentionally simple and model-agnostic so that it can be easily integrated into a wide range of 3D editing systems. Despite this simplicity, the performance gains come not from adding more views but from how these views are selected. Our 3D Gaussian Novel View Sampling is the core contribution: it probabilistically targets unexplored regions, avoids redundant viewpoints through iterative updates, and selects only a small number of informative novel views, leading to clear improvements in unseen-view quality. (2) To the best of our knowledge, no prior work addresses novel-view selection in 3D editing; existing methods rely only on reference views and ignore unseen-view consistency. While simple alternatives such as random or uniform sampling are possible, we include them as baselines, and Table 14 shows that both are consistently outperformed by 3DG-PVS. This confirms that our gains come from a principled sampling strategy rather than merely adding more views.
>
> [**Q3**] 3DG-PVS can be a minor contribution.
>
> [**A3**] 3DG-PVS is not a minor contribution. It introduces a probabilistic and iterative view-sampling algorithm specifically designed for the 3D editing setting, where novel-view degradation has not been previously studied or addressed. Unlike simple heuristics such as random or uniform sampling, 3DG-PVS models reference-view coverage, targets unexplored regions via inverse-probability sampling, avoids redundancy through iterative updates, and removes implausible viewpoints with outlier filtering. As shown in the ablations in Table 2, each component is essential, and the full strategy consistently outperforms random and uniform sampling with far fewer views. This makes 3DG-PVS a key technical contribution rather than a minor addition.
>
> [**Q4**] (1) The method is an unfair comparison. In general, more views for optimisation should have better performance. (2) Even the Regular strategy might have better performance when combined with the baseline methods.
>
> [**A4**] (1) Our comparison is fair. The additional views used by our method are not extra ground-truth reference images, but unseen views rendered from the same 3D structure and then enhanced using diffusion editing. Thus, we do not use more real data than the baselines. To further ensure fairness, we also perform a strict equal-time comparison where our method is given exactly the same training time as each baseline. As shown in Table 15, even under this controlled setting, our method consistently outperforms all baselines, demonstrating that the gains arise from effective view selection rather than from using more views. (2) We agree that even a simple regular sampling strategy can improve editing quality, but this only reinforces the importance of choosing informative novel views. Our 3DG-PVS maximizes this effect by targeting unexplored regions, avoiding redundancy, and removing outliers. As shown in Table 2 and Table 14, it consistently outperforms other novel view samplings, confirming its necessity and contribution.
>
> [**Q5**] Why does the number differ so much (0.22 VS. 0.78) in human evaluation ?
>
> [**A5**] Our human evaluation involved 50 participants comparing baseline vs. baseline+NVE-Adaptor on 40 seen and 40 unseen views per scene. The large gap (0.22 vs. 0.78) is primarily due to the unseen views: while seen views show a moderate preference (0.41 vs. 0.59), unseen views show a much stronger preference for our method (0.03 vs. 0.97). The final score reflects the aggregation of both categories.
>
> [**Q6**] In Table 2, T*=40 is better than T*=30.
>
> [**A6**] The 0.641 in T*=30 is a typo, 0.651 is correct.

---

### Official Review · Reviewer_hNRr · 2025-10-31

**Soundness:** 3
**Presentation:** 3
**Contribution:** 3
**Rating:** 4
**Confidence:** 5

**Summary:**

This paper brought up an interesting challenge of limited view coverage in current 3D editing, where the edited 3D model exhibits visual artifacts in the views that are distant from the edited reference images. To address this issue, the authors propose a novel view sampling strategy to expand the reference views. Subsequently, the newly sampled views are forwarded into multi-view diffusion model along with original reference views for editing. Experiments show the expansion achieves significant improvement on the unseen view’s visual quality.

**Strengths:**

1.This paper discusses on a novel and practical view coverage issue in current 3D editing task. Starting from this challenge, the author propose a novel view sampling strategy that can be plug-and-played by current 3D editing methods.

2.Both qualitative and quantitative experiments in this paper are comprehensive and reasonably designed. The presented visual quality of the rendered video results are promising.

3.The paper is clearly presented and easy to follow

**Weaknesses:**

My concerning mainly lies on the setting. In my understanding, the main contribution of this paper is the view sampling issue. In another world, after optimally selecting the novel views, the rest processing is feed-forwarding the views to current 3D editing models. Current experiments seem to focus more on the comparison between the results with view expansion and that without view expansion to emphasis the effect of view expansion. In my opinion, the idea of
using view expansion itself is not sufficiently novel, as it is a common sense in reconstruction. Instead, I would like to see the improvement of proposed view sampling strategy corresponding to the baseline random sampling or uniform sampling, presenting how and why the proposed sampling strategy outperforms a vanilla strategy. I would temporarily give a borderline reject rating and raise my score upon this contribution is well illustrated.

**Questions:**

See weaknesses. I would suggest the authors respond to the concern.

---

> ### Author Response · Authors · 2025-12-02
>
> [**Q1**] In my opinion, the idea of using view expansion itself is not sufficiently novel, as it is a common sense in reconstruction. Instead, I would like to see the improvement of proposed view sampling strategy corresponding to the baseline random sampling or uniform sampling, presenting how and why the proposed sampling strategy outperforms a vanilla strategy. I would temporarily give a borderline reject rating and raise my score upon this contribution is well illustrated.
>
> [**A1**] We fully agree that the core contribution of this work lies in the design of the view sampling strategy rather than the concept of view expansion itself. To demonstrate why our proposed method outperforms vanilla strategies, we first defined four essential criteria for effective novel view selection:
>
> (1) Unexplored Region Coverage: The sampling must prioritize regions not covered by the reference views.
>
> (2) Novel View Sparsity: Selected novel views should be spatially distinct (sparse) to maximize information gain and avoid redundancy.
>
> (3) Efficiency: The method must maximize surface coverage with the minimum number of views to maintain computational efficiency.
>
> (4) Outlier Handling: It must filter out geometrically implausible views (e.g., too far or too close) to ensure stability.
>
> Based on these criteria, our 3D Gaussian Probabilistic View Sampling (3DG-PVS) is designed to satisfy all four conditions, whereas baseline strategies fail to meet them simultaneously.
>
> 1. Why 3DG-PVS outperforms baselines
>
> Proposed 3DG-PVS: We construct a Gaussian probability distribution based on reference views and sample from its inverse to target unexplored regions (Criteria 1). By employing iterative sampling, we continuously update the probability map to prevent clustering, ensuring high sparsity (Criteria 2). Furthermore, we incorporate an outlier removal mechanism to constrain the search space to a plausible range (Criteria 4) and experimentally determined the optimal view count for maximum efficiency (Criteria 3).
>
> 2. Comparison with Baselines
>
> **vs. Random Sampling**: Random sampling is inherently unstable. It offers no guarantee for covering unexplored regions (failing Criteria 1) and often results in clustered views (failing Criteria 2). Moreover, without outlier handling (failing Criteria 4), it creates significant variance in reconstruction quality, making it difficult to determine an optimal (failing Criteria 3), stable view count.
>
> **vs. Uniform Sampling**: While uniform sampling can address coverage and sparsity (Criteria 1 & 2), it is highly inefficient. It requires a significantly larger number of views to achieve comparable coverage, which drastically increases training time and computational cost (failing Criteria 3). Additionally, it lacks context-awareness regarding the object's scale, leading to poor outlier handling (failing Criteria 4). Furthermore, quantitative and qualitative comparisons with Uniform sampling are presented in Figure 7 and Table 2.
>
> To address your concern directly, we have summarized this comparative analysis in Table A. We reflect this as Table 14 into our revision.
>
> Table A. Quantitative comparison with random and uniform sampling strategies. We report the average scores across four baselines (IN2N, GCtrl, VcEdit, and DGE). Note that for uniform sampling, we applied our outlier removal scheme to ensure computational tractability; otherwise, the novel view count would be intractable. For random sampling and our sampling, the number of views is fixed at 30.
>
> ||unexplored region|novel view sparsity|efficiency|outlier handling|FID (seen view)|SSIM (seen view)|FID (unseen view)|SSIM (unseen view)|Training Time|
> |--|--|--|--|--|--|--|--|--|--|
> |Random|X|X|X|X|236|0.654|286|0.598|36 min|
> |Uniform|O|O|X|X|217|0.688|264|0.610|124 min|
> |3DG-PVS|O|O|O|O|212|0.693|230|0.651|35 min|

---

### Official Review · Reviewer_syaU · 2025-10-31

**Soundness:** 3
**Presentation:** 3
**Contribution:** 3
**Rating:** 4
**Confidence:** 3

**Summary:**

The paper targets the unseen-view inconsistency problem in multi-view 3D editing: edits look good near trained reference views but degrade at novel viewpoints. It proposes NVE-Adaptor, which (i) samples novel viewpoints via a probabilistic strategy (3DG-PVS), (ii) renders those views, (iii) enhances them with diffusion-based multi-image editing (with prompts like “high quality”), and (iv) uses the edited renders as extra supervision during 3D editing, improving view consistency on Mip-NeRF 360 and IN2N, plus a real-world set.

**Strengths:**

- The proposed NVE-Adaptor only augments supervision with edited novel-view images and can attach to various NeRF / 3DGS edit pipelines.

- Across several baselines (IN2N, GaussCtrl, VcEdit, DGE), the adaptor improves FID/SSIM and CLIP-based consistency on both seen and unseen views.

**Weaknesses:**

- The method improves novel-view renders by editing them with a text-to-image model (multi-image consistency + prompts like “high quality”) before using them as supervision. This can hallucinate textures/geometry, drifting from the true scene and biasing the 3D optimization. Therefore, it would be good to quantify drift with image-space metrics to ground-truth photos where available (you already capture real images + COLMAP extrinsics for the unseen set) and report an edit-intensity control or mask-based constraints to limit semantic changes.

- Tables report average scores, but variances / CIs are missing. Several gains (e.g., CLIP/SSIM deltas) are modest. Add std/CI over seeds (you mention 10 or 30 seeds in places) to support significance claims, especially for HumanFID and directional CLIP where variance can be high.

- 3DG-PVS uses isotropic Σ with σ=0.4 and a cube-based outlier-removal heuristic. The performance sensitivity to these hyper-parameters is unclear. It would be good to add a comprehensive hyperparameter analysis.

- The authors provided per-scene training time overheads (e.g., IN2N: 56→68 min; DGE: 15→18 min), but not inference-time cost or total GPU hours for the full training + novel-view editing pipeline. Please add wall-clock and GPU memory profiles for typical scenes, and break down where time is spent (rendering vs diffusion editing vs 3D optimization). This would help better understand the compute-quality trade-offs.

- Table 8 shows that when reference views are very sparse (N≤40), gains shrink or vanish; at N=20 performance can even degrade (Base vs Base+NVE nearly equal or worse). Please discuss failure modes (diffusion editor invents content due to under-constrained views) and consider confidence-based filtering to discard low-reliability edited novel views in those regimes.

**Questions:**

Please refer to the weakness section.

---

> ### Author Response · Authors · 2025-12-02
>
> [**Q1**] (1) The method can hallucinate textures/geometry, drifting from the true scene and biasing the 3D optimization. Therefore, (2) it would be good to quantify drift with image-space metrics to ground-truth photos available.
>
> [**A1**] (1) To mitigate this issue, our method is explicitly designed to condition the diffusion model on both the reference image and the edited novel-view image, as described in Lines 260–261. By concatenating these inputs, the diffusion model is guided to preserve scene-consistent structures while still applying the desired semantic edits.
> (2) To further validate this design choice, we conducted an ablation study about 3D reconstruction that quantitatively measures the impacts of drift using ground-truth real images. Specifically, for the unseen-view sets, we used the captured real images as ground truth and rendered predictions from the reconstructed 3D structures using the corresponding COLMAP extrinsics. We then tested several input configurations for the diffusion model:
>
>  (A) reference views only
>
>  (B) reference views and edited novel views processed independently
>
>  (C) concatenated reference and edited novel views at a 1:1 ratio
>
>  (D) concatenated reference and edited novel views at a 1:2 ratio
>
>  (E) concatenated reference and edited novel views at a 2:1 ratio
>
>  (F) concatenated reference and edited novel views at a 3:1 ratio
>
>  (G) concatenated reference and edited novel views at a 4:1 ratio
>
> The results reported in Table 11 of our Appendix show that configuration (B)—processing reference and edited novel views independently—indeed increases drift compared to (A), confirming the reviewer’s concern. In contrast, all configurations (C-G) that jointly condition the diffusion model on both reference and edited novel-view images by concatenation significantly reduce drift and improve view consistency.
>
> [**Q2**] Tables report average scores, but confidence intervals are missing.
>
> [**A2**] We provide detailed analysis of Table 1 into Table 4 and Table 5 in terms of mean and 95% confidence interval in Appendix E. We also updated Tables 6, 7, and 8 to contain the mean and 95% confidence interval. These additional statistics confirm that the observed performance gains remain statistically significant.
>
> [**Q3**] 3DG-PVS uses isotropic Σ with σ=0.4 The performance sensitivity to these hyper-parameters is unclear. It would be good to add a comprehensive hyperparameter analysis.
>
> [**A3**] Table 12 of our Appendix provides a sensitivity analysis with respect to the Gaussian variance parameter σ. Our results show a stable performance region within 0.2<σ<0.7, indicating that the method is robust to moderate changes in σ.
>
> [**Q4**] Please add wall-clock and GPU memory profiles for typical scenes, and break down where time is spent (rendering vs diffusion editing vs 3D optimization).
>
> [**A4**] We have added Table 13 of our Appendix, which reports detailed time and memory profiles for each baseline model before and after applying our method. For training, we break down the wall-clock time into three components: diffusion-based editing, rendering, and 3D optimization. For inference, we additionally report the novel-view rendering speed (fps) for typical scenes. We also provide GPU memory statistics, including total GPU hours and peak memory usage for the full training pipeline, as well as the GPU hours and memory consumption specifically attributable to the diffusion editing stage.
>
> [**Q5**] Table 8 shows that when reference views are very sparse (N≤40), gains shrink or vanish; at N=20 performance can even degrade. (1) Please discuss failure modes and (2) consider confidence-based filtering to discard low-reliability edited novel views in those regimes.
>
> [**A5**] (1) As discussed around Line 776, when the number of reference views is very small, the 3D reconstruction becomes severely under-constrained and the unseen-view renderings become overly blurry and semantically ambiguous. In this regime, the diffusion model struggles to interpret the content of the rendered images and no longer behaves as a faithful editor of the underlying scene. Instead, it tends to hallucinate new content that is not well aligned with the true geometry or appearance, which explains why the gains shrink or even vanish at N=20 in Table 8.
>
> (2) In Appendix C, we are proposing an outlier-removal module as filtering low-reliable viewpoints. To be specific, we construct a 3D region (with a small margin) covered by the reference viewpoints and treat this as a decision boundary: novel viewpoints inside this region are assigned a confidence score of 1, while those outside are assigned 0 and discarded before being used as novel views. As shown in Table 2, this outlier removal improves robustness and helps avoid low-reliability novel viewpoints.

---

### Official Review · Reviewer_NoU9 · 2025-11-01

**Soundness:** 3
**Presentation:** 2
**Contribution:** 2
**Rating:** 4
**Confidence:** 4

**Summary:**

- This paper proposes NVE-Adaptor, a plug-and-play module that improves multi-view consistent 3D editing, especially for unseen viewpoints that were not part of the original reference views.

- The method explores novel camera viewpoints, renders images from those new views, refines them with diffusion-based editing, and uses them as extra supervision to improve consistency in 3D editing results.

- This paper works in a model-agnostic manner, complementing existing 3D editing pipelines (e.g., NeRF / Gaussian Splatting–based systems), and shows consistent improvements on seen and unseen views across multiple benchmarks and real-world scenarios.

**Strengths:**

- This paper clearly identifies a critical limitation of existing multi-view 3D editing systems, degradation and inconsistency when rendering from unseen viewpoints, and provides quantitative sensitivity analysis to motivate the problem.

- This paper proposes a simple, model-agnostic adaptor that can be plugged into existing 3D editing pipelines without architectural changes, making it widely applicable to NeRF-based and Gaussian Splatting-based systems.

- This paper provides strong empirical evidence across multiple datasets, demonstrating consistent quality improvements in both seen and unseen viewpoints, including real-world data, validating the practicality and robustness of the method.

**Weaknesses:**

- In demo.mp4 around the timestamp 00:58, most of the input 3D characteristics disappear during the editing process. This phenomenon appears in multiple examples, yet it is not addressed in the limitations section, which is disappointing.

- The illustration in Figure 3 describing the functionality of the Novel View Editing-Adaptor (NVE-Adaptor) could be made more intuitive. Instead of using red and blue dots, the figure could be revised so that readers can clearly distinguish reference views and novel views without referring back to a glossary.

- Including the code snippet (source_code.py in supplementary) in the supplementary material is appreciated, but the implementation details are too brief. To ensure reproducibility, more detailed explanations should be provided, and relevant code components should be explicitly linked or referenced in the appendix.

- For experimental settings such as Table 8, where specific parameters are used, it would be helpful to explain why those parameter choices were made. Providing this rationale would improve the clarity and interpretability of the paper.

**Questions:**

Mentioned in the weaknesses.

---

> ### Author Response · Authors · 2025-12-02
>
> [**Q1**] In demo.mp4 around the timestamp 00:58, most of the input 3D characteristics disappear during the editing process.
>
> [**A1**] We would like to clarify that the disappearance of certain 3D characteristics in the demo (e.g., the removal of the beard at 00:54 - 00:59) is not a failure of our method, but the intended result of the semantic edit requested by the target text (“Turn it into batman”). In the example shown at that timestamp, the instruction is to turn the male subject into Batman. During this transformation, facial attributes such as the beard are removed because the pretrained diffusion model’s learned representation of “Batman” typically does not include a beard. Thus, these changes reflect the semantics of the target prompt, rather than unintended loss of geometry. We also note that what may appear to be “preserved beard” in the baseline is actually needle-like noise caused by poor unseen-view reconstruction, not meaningful beard. Our method removes these artifacts while performing semantic editing aligned with the text prompt. More broadly, controllability in semantic editing is inherently constrained by the pretrained diffusion model’s internal biases and learned distributions. This is a fundamental limitation of pre-trained diffusion model itself, not a limitation specific to our NVE-Adaptor.
>
> [**Q2**] The illustration in Figure 3 describing the functionality of the Novel View Editing-Adaptor (NVE-Adaptor) could be made more intuitive. Instead of using red and blue dots, the figure could be revised so that readers can clearly distinguish reference views and novel views without referring back to a glossary.
>
> [**A2**] We have updated Figure 3 so that reference views and novel views are more intuitively distinguishable by using different shapes and colors, and by adding explicit labels within the figure. We also applied the same visual conventions consistently across Figures 4 and 7 to ensure clarity and coherence throughout the paper.
>
> [**Q3**] Including the code snippet (source_code.py in supplementary) in the supplementary material is appreciated, but the implementation details are too brief. To ensure reproducibility, (1) more detailed explanations should be provided, and (2) relevant code components should be explicitly linked or referenced in the appendix.
>
> [**A3**] (1) We added a more detailed explanation about our sampling process as another section of implementation details in Appendix D. (2) We also revised the supplementary source code by adding more detailed comments to explain the implementation. In addition, we included explicit references linking the relevant code components to the corresponding equations in the paper, ensuring that the supplementary materials and the main text are mutually connected.
>
> [**Q4**] For experimental settings such as Table 8, where specific parameters are used, it would be helpful to explain why those parameter choices were made. Providing this rationale would improve the clarity and interpretability of the paper.
>
> [**A4**] As discussed in Lines 772–778, the parameter choices (i.e., specifically reducing the number of reference viewpoints from 100 down to 20) in Table 8 were designed to evaluate how sensitive our NVE-Adaptor is to varying levels of reference-view sparsity. We selected these values to gradually decrease the number of available reference views and observe the corresponding impact on performance with and without our method. The step size of 20 was chosen because, based on our preliminary analysis with the base model, a reduction of 20 reference views was the minimum decrement that produced noticeable changes across all evaluation metrics (FID, CLIP, LPIPS, SSIM). Therefore, we adopted 20 as a meaningful and interpretable interval for sparsity analysis. We will clarify this rationale in the revised version (Line 774 - 775) of the paper.

---

### Meta-Review · Area_Chair_V9AC · 2025-12-23

**Summary:**

The paper proposes NVE-Adaptor, a method designed to enhance the rendering quality of unseen viewpoints in 3D editing tasks. The authors introduce a 3DG-PVS strategy to sample novel views, which are then edited by a diffusion model and used as additional supervision.

After carefully considering the reviews and the rebuttal, I recommend Rejection. While the authors have presented a practical solution and provided a solid rebuttal, all reviewers assigned scores leaning towards the negative side (Reject or Borderline Reject). The consensus reflects a shared concern regarding the scope of the contribution. Specifically, as noted by Reviewer CjWi, the core premise—that adding supervision from novel views improves their quality—is largely considered an expected behavior or common sense in 3D optimization. While the proposed sampling strategy is effective, the overall technical novelty is viewed as incremental for a top-tier venue.

**Reviewer Concerns:**

**Addressed Concerns:**

1. The authors thoughtfully addressed requests for baselines by comparing their method against Random and Uniform sampling.

2. Concerns regarding semantic drift were well-handled with additional ablation studies in the rebuttal.

**Outstanding Concerns (Key Factors for Decision):**

1. Novelty and Contribution (Reviewer CjWi and hNRr): The primary reservation is that addressing the degradation of unseen views by simply "adding more views" falls into the realm of common sense or standard practice. While the proposed sampling heuristic (3DG-PVS) is shown to be better than random sampling.
2. Significance: The paper focuses on how to sample views to add to the training set. While useful, the committee views this as an incremental refinement, which aligns with the overall low scores received.

**Reviewer Scores:**

Reviewer CjWi (Score: 2): Maintained the view that while the phenomenon is visualized well, the solution is "very simple" and the improvement is expected when using more views.

Reviewer hNRr (Score: 4): Acknowledged the practicality but also noted that the concept of view expansion itself is not sufficiently novel.

Other Reviewers (Score: 4): Found the method sound but did not champion the paper for acceptance.

The unanimous lack of strong support (no scores above 4) confirms that the paper is currently below the acceptance threshold.

---

### Decision · Program_Chairs · 2026-01-26

Reject